# Same-Cluster Querying for Overlapping Clusters

**Wasim Huleihel**
Department of Electrical Engineering
Tel-Aviv University
Tel-Aviv, Israel 6997801
`wasimh@mail.tau.ac.il`

**Arya Mazumdar**
College of Information & Computer Sciences
University of Massachusetts Amherst
Amherst, MA 01003
`arya@cs.umass.edu`

**Muriel Médard**
Electrical Engineering & Computer Science
Massachusetts Institute of Technology
Cambridge, MA 02139
`medard@mit.edu`

**Soumyabrata Pal**
College of Information & Computer Sciences
University of Massachusetts Amherst
Amherst, MA 01003
`soumyabratap@umass.edu`

## Abstract

Overlapping clusters are common in models of many practical data-segmentation applications. Suppose we are given $n$ elements to be clustered into $k$ possibly overlapping clusters, and an oracle that can interactively answer queries of the form "do elements $u$ and $v$ belong to the same cluster?" The goal is to recover the clusters with minimum number of such queries. This problem has been of recent interest for the case of disjoint clusters. In this paper, we look at the more practical scenario of overlapping clusters, and provide upper bounds (with algorithms) on the sufficient number of queries. We provide algorithmic results under both arbitrary (worst-case) and statistical modeling assumptions. Our algorithms are parameter free, efficient, and work in the presence of random noise. We also derive information-theoretic lower bounds on the number of queries needed, proving that our algorithms are order optimal. Finally, we test our algorithms over both synthetic and real-world data, showing their practicality and effectiveness.

## 1   Introduction

Recently, semi-supervised models of clustering that allow active querying during data segmentation have become quite popular. This includes active learning, as well as data labeling by amateurs via crowdsourcing. Clever implementation of interactive querying framework can improve the accuracy of clustering and help in inferring labels of large amount of data by issuing only a small number of queries. Interactions can easily be implemented (e.g., via captcha), especially if queries involve few data points, like pairwise queries of whether two points belong to the same cluster or not [1, 2, 4, 9, 13, 15, 18, 19, 21, 22, 23, 25].

Until now, the querying model and algorithms/lower bounds are highly tailored towards flat clustering that produces a partition of the data. Consider the problem of clustering from pairwise queries such as above when an element can be part of multiple clusters. Such overlapping clustering instances are ubiquitous across areas and many time are more practical model of data segmentation, see [3, 5, 20, 26]. Indeed, overlapping models are quite natural for communities in social networks or topic models [16]. In the supervised version of the problem every element (or data-point) can have multiple labels, and we would like to know all the labels. To see how the querying might work here consider the following input: {Tiger Shark, Grizzly Bear, Blue Whale, Bush Dog, Giant Octopus, Ostrich, Komodo Dragon}. This set

can be clustered into the mammals {Grizzly Bear, Blue Whale, Bush Dog}, marine-life {Tiger Shark, Blue Whale, Giant Octopus}, non-mammals {Tiger Shark, Giant Octopus, Ostrich, Komodo Dragon}, land-dwellers {Grizzly Bear, Bush Dog, Ostrich, Komodo Dragon}. Quite clearly, this ideal clustering (without labels) is overlapping. If a query of whether two elements belong to the same cluster or not is made then the answer should be 'yes' if there exists a cluster where they can appear together. If we form a response matrix of size $7 \times 7$ with rows and columns indexed by the order they appeared above in the list and entries being the binary answers to queries, then the matrix would be following:

$$
\begin{bmatrix}
 & \text{TS} & \text{GB} & \text{BW} & \text{BD} & \text{GO} & \text{Os} & \text{KD} \\
\text{TS} & * & 0 & 1 & 0 & 1 & 1 & 1 \\
\text{GB} & 0 & * & 1 & 1 & 0 & 1 & 1 \\
\text{BW} & 1 & 1 & * & 1 & 1 & 0 & 0 \\
\text{BD} & 0 & 1 & 1 & * & 0 & 1 & 1 \\
\text{GO} & 1 & 0 & 1 & 0 & * & 1 & 1 \\
\text{Os} & 1 & 1 & 0 & 1 & 1 & * & 1 \\
\text{KD} & 1 & 1 & 0 & 1 & 1 & 1 & *
\end{bmatrix}
$$

What is the minimum number of adaptive queries that we should make to the above matrix so that it is possible to recover the clusters? In the case when the clusters are not overlapping, the answer is $nk$, where $n$ is the number of elements and $k$ is the number of possible clusters [19]. For the case of overlapping clusters it is not clear whether there is a unique clustering that explains the responses. For this, certain extra constraints must be placed: for example, a reasonable assumption is that an element can only be part of $\Delta$ clusters, among the total $k$ possible clusters, $\Delta \ll k$.

Just like the response matrix above, it is possible to form a *similarity matrix*. The $(i, j)$th entry of this matrix simply is the number of clusters where the $i$th and the $j$th elements coexists. It is clear that the response matrix is just a quantized version of the similarity matrix. Even when the entire similarity matrix is given, there is no guarantee on uniqueness of overlapping clustering, unless further assumptions are made. In this paper, we primarily aim to recover the clustering from a limited number of adaptive queries to the response matrix. However, in terms of uniqueness guarantees, we often have to stop at the uniqueness of the similarity matrix.

**Main results and Techniques.** Recovery of overlapping clusters from budgeted same-cluster queries is widely open, and significantly more challenging than the flat clustering counterpart. In fact, none of the techniques proposed in prior literature as mentioned above extends easily to the case when the clusters may overlap. In this paper we tackle this problem for various types of responses. Specifically, in our setting there is an oracle having access to the similarity matrix $\mathbf{A}\mathbf{A}^T$, where $\mathbf{A}$ is the $n \times k$ clustering matrix whose $i$th row is the indicator vector of the cluster membership of $i$th element. In its most powerful mode, when queried the oracle provides the number of clusters where $i$th and the $j$th elements coexists, namely the values of the entries of the matrix $\mathbf{A}\mathbf{A}^T$. It turns out, however, that even if one knows the matrix $\mathbf{A}\mathbf{A}^T$ perfectly, it is not always possible to recover $\mathbf{A}$ up to a permutation of columns.[1] In fact, if no assumption is imposed on the clustering matrix $\mathbf{A}$, then recovering $\mathbf{A}$ from $\mathbf{A}\mathbf{A}^T$ is intractable in general. Indeed, even just finding conditions on $\mathbf{A}$ such that the factorization is unique (up to permutations) is related to the famous unsolved question of *"for which orders a finite projective plane exists?"* [11]. It is then clear that we need to impose some assumptions. We tackle this inherent problem in two different approaches. First, in Sections 3.1 and 3.2, we propose two *generative* models for $\mathbf{A}$: 1) a uniform ensemble where a given element can only be part of $\Delta \geq 1$ clusters,[2] among the total of $k$ clusters, and its membership is drawn uniformly at random among all possible $\binom{k}{\Delta}$ possible placements, 2) the matrix $\mathbf{A}$ is generated i.i.d. with Bernoulli entries. Then, for these two ensembles, we investigate the above fundamental question, and derive sufficient conditions under which this factorization is unique, along with quasi-polynomial worst-case complexity algorithms for recovering $\mathbf{A}$ from $\mathbf{A}\mathbf{A}^T$. The main immediate implication of this result is that, under certain conditions, the clustering recovery problem reduces to recovering the similarity matrix, placing this objective to be our main task.

While the above generative models allow us to obtain elegant and neat theoretical results, one might argue that they may not capture many challenges existing in real-world overlapping clustering problems. To this end, in Section 3.3, we go beyond the above generative models and analyze a

general worst-case model with no statistical assumptions. Then, under certain realistic assumptions on the clustering matrix, we provide and analyze algorithms solving the recovery problem.

In practice, however, the aforementioned 'value' oracle responses might be quite expensive. Accordingly, we study also quantized and noisy variants of these responses. For example, instead of getting direct values from $\mathbf{A}\mathbf{A}^T$, the oracle only supplies the learner with (possibly noisy) binary answers on whether arbitrarily picked pair of elements $(i, j)$ appear together in some cluster or not ('same-cluster query'). We consider also the case of dithered oracle, where noise is injected before quantization. For these scenarios (and others), we provide both lower and upper bounds on the number of queries needed for exact recovery. Our lower bounds are obtained using standard information-theoretic results, such as Fano's inequality. For the upper bounds, we design novel randomized algorithms for recovering the similarity matrix, and further show that these algorithms can work when the noise parameter is not given in advance. For example, when $k = O(\log n)$ and $\Delta \ll k$, we show that the sufficient and necessary number of quantized queries is $\Theta(n \log n)$, for the uniform ensemble. Finally, we test our algorithms over both synthetic and real-world data, showing the practicality and effectiveness of our algorithms.

**Related Work.** As mentioned above, there is a series of applied and theoretical works studying the query complexity of 'same-cluster' queries for objective-based clustering (such as, $k$-means) and clustering with statistical generative models. In all the cases though, the clusters are assumed to be non-overlapping. From a practical standpoint, entity resolution via crowdsourced pairwise same-entity queries were studied in [13, 17, 24, 25, 22, 27]. The effect of (possibly noisy) 'same-cluster' queries in similarity matrix based clustering has been studied in [18, 19, 1]. On the other hand, the effect of 'same-cluster' queries in the efficiency of $k$-means clustering was initiated in [4] - which was then subsequently further studied in [27, 2, 9].

In our approach, we crucially use the 'low-rank' structure of the similarity matrix to recover the clustering from a bounded number of responses. Low-rank matrix completion is a well-studied topic in statistics and data science [7, 14]. It is possible to obtain weaker version of some of our results by relying on the results of low-rank matrix completion as black-boxes. However, the specific structure of the similarity matrix under consideration allows us to obtain stronger results. The response matrix is a quantized version of a low-rank matrix. Querying entries of the response matrix can be seen as a so called 1-bit matrix completion problem [12]. However, most of the recovery guarantees of 1-bit matrix completion depends crucially on certain *dither* noise (see, [12]), which may not be what is allowed in our setting. Finally, we mention [6] where the problem of overlapping clustering was considered from a different point of view.

**Organization.** The remaining part of this paper is organized as follows. The model and the learning problem are provided in Section 2. Our main results on the query complexity are presented in Section 3. In particular, we provide upper bounds (with algorithms) on the sufficient number of queries for each of the scenarios investigated in this paper. These results are also accompanied with information-theoretic lower bounds on the necessary number of queries, which are presented in the appendix due to page limitation. Finally, Section 4 is devoted for a numerical study, where our main results are illustrated empirically. Detailed proofs of all the theoretical results can be found in the supplementary material.

## 2 Model and Learning Problem

Consider a set of elements $\mathcal{N} \equiv [n]$ with $k$ latent clusters $\mathcal{N}_i$, for $i = 1, 2, \ldots, k$, such that each element in $\mathcal{N}$ belongs to at least one cluster. This data is represented by an $n \times k$ matrix $\mathbf{A}$, where $\mathbf{A}_{i,j} = 1$ if the $i$th elements is in the $j$th cluster. We will denote the $k$-dimensional binary vector representing the cluster membership of the $i$th element by $\mathbf{A}_i$ (i.e., the $i$th row of $\mathbf{A}$), and will henceforth refer to it as the $i$th *membership vector*. In our setting there is an *oracle* $\mathcal{O} : \mathcal{N} \times \mathcal{N} \to \mathcal{D}$ that when queried with a pair of elements $(i, j) \in \mathcal{N} \times \mathcal{N}$, returns a natural number $L \in \mathcal{D} \subset \mathbb{N}$ according to some pre-defined rule. We shall refer to $\mathcal{O}$ as the *oracle map*. The queries $\Omega \subseteq \mathcal{N} \times \mathcal{N}$ can be done adaptively. Our goal is to find the set $\Omega \subseteq \mathcal{N} \times \mathcal{N}$ such that $|\Omega|$ is minimum, and it is possible to recover $\{\mathcal{N}_i\}_{i=1}^k$ from the oracle answers. More specifically, the oracle have access to the similarity matrix $\mathbf{A}\mathbf{A}^T$, and when queried with $\Omega$, answers according to $\mathcal{O}$. Given $(i, j) \in \mathcal{N} \times \mathcal{N}$, we consider the following oracle maps $\mathcal{O}$, capturing several aspects of the problem:

- *Direct responses*: The oracle response is $\mathcal{O}_{\mathsf{direct}}(i,j) = \mathbf{A}_i^T \mathbf{A}_j$, namely, the number of clusters that elements $i$ and $j$ belong to simultaneously. Note that when the clusters are disjoint, the output is simply an answer to the question "do elements $i$ and $j$ belong to the same cluster?".
- *Quantized (noisy) responses*: The oracle response is $\mathcal{O}_{\mathsf{quantized}}(i,j) = \mathcal{Q}\left(\mathbf{A}_i^T \mathbf{A}_j\right) \oplus W_{i,j}$, where $\mathcal{Q}(x) \triangleq 1$ for $x > 0$, and 0, otherwise, and $W_{ij} \sim \mathsf{Bernoulli}(q)$, with $0 \leq q \leq 1$, independent over pairs $(i,j)$. In the noiseless case, i.e., $q = 0$, the oracle response is whether elements $i$ and $j$ appears together in at least one cluster or not. In the noisy case, the oracle response is the quantized response with probability $1 - q$, and flipped with probability $q$. This can be interpreted as if the quantized responses are further sent through a binary symmetric channel $\mathsf{BSC}(q)$.
- *Dithered responses*: The oracle response is $\mathcal{O}_{\mathsf{dithered}}(i,j) = \mathcal{Q}\left(\mathbf{A}_i^T \mathbf{A}_j + Z_{i,j}\right)$, where $Z_{ij} \sim \mathsf{Normal}(0, \sigma^2)$, independent over pairs $(i,j)$. In other words, the oracle outputs a dithered and quantized version of the direct responses.

For simplicity of notation, throughout the rest of this paper we will denote the oracle response to query $(i,j) \in \mathcal{N}$ by $\mathbf{Y}_{ij}$, irrespective of the oracle model, which will be clear from the context.

If we permute the columns of $\mathbf{A}$, the gram matrix $\mathbf{A}\mathbf{A}^T$ will remain the same. Therefore, finding $\mathbf{A}$ is only possible up to a permutation of columns. Unfortunately, it turns out that even if we know $\mathbf{A}\mathbf{A}^T$ perfectly it is not always possible to find $\mathbf{A}$ up to a permutation of columns, namely, the factorization may not be unique. As an example, consider the following matrices

$$
\begin{bmatrix} 1 & 1 & 0 & 0 \\ 0 & 1 & 1 & 0 \\ 0 & 0 & 1 & 1 \\ 1 & 0 & 1 & 0 \end{bmatrix} \quad \text{and} \quad \begin{bmatrix} 0 & 0 & 1 & 1 \\ 0 & 1 & 0 & 1 \\ 1 & 1 & 0 & 0 \\ 1 & 0 & 0 & 1 \end{bmatrix}
$$

which have the same gram matrix but evidently are not column permutations of each other. Hence, even if we observe all the entries of the gram matrix, it is not possible to distinguish between these two matrices. We tackle this inherent problem in two different approaches.

**Generative Models.** We consider two generative models (random ensembles) for $\mathbf{A}$; *uniform* and *i.i.d.* ensembles, defined as follows. Given $k, \Delta \in \mathbb{N}$, define the set

$$
T_k(\Delta) \triangleq \{\mathbf{c} \in \{0,1\}^k : \mathsf{w}_{\mathsf{H}}(\mathbf{c}) = \Delta\}, \tag{1}
$$

as the set of all $k$-length binary sequence with Hamming weight ($\mathsf{w}_{\mathsf{H}}$) $\Delta$. Then, we say that $\mathbf{A}$ belongs to the uniform ensemble if $\mathbf{A}$ is formed by drawing independently its $n$ rows from $T_k(\Delta)$. In the latter ensemble, the matrix $\mathbf{A}$ is an i.i.d. matrix, with each entry being a $\mathsf{Bernoulli}(p)$ random variable, where $0 \leq p \leq 1$. As mentioned above we are interested in exact recovery of the clusters $\{\mathcal{N}\}_{i=1}^k$, or equivalently, the clustering matrix $\mathbf{A}$ up to a permutation of the columns of $\mathbf{A}$. More precisely, we define the average probability of error associated with an algorithm which outputs an estimate $\hat{\mathbf{A}}$ of $\mathbf{A}$ by $\mathsf{P}_{\mathsf{error}} \triangleq \mathbb{P}\left(\bigcap_{\pi \in \mathbb{S}_k} \{\hat{\mathbf{A}} \neq \mathbf{A}\mathbf{P}_\pi\}\right)$, where $\mathbf{P}_\pi$ is the permutation matrix corresponding to the permutation $\pi : [k] \to [k]$, and $\mathbb{S}_k$ is the symmetric group acting on $[k]$. Accordingly, we say that an algorithm properly recovered $\mathbf{A}$ if $\mathsf{P}_{\mathsf{error}}$ is small. This recovery criterion follows from the fact that clustering is invariant to a permutation of the labels.

In contrast to the above negative example where two different matrices have the same gram matrix, under certain weak conditions presented in Appendix A (see, Lemmas 3 and 4), we show that if the matrix $\mathbf{A}$ is generated according to the either one of the above random ensembles, the factorization is unique up to column permutations. This results have a straightforward implication: under the conditions of the Lemmas 3 and 4, the clustering recovery problem (i.e., recovering $\mathbf{A}$) reduces to the recovery of the similarity matrix $\mathbf{A}\mathbf{A}^T$, given partial observations $\Omega$ of its entries through the oracle $\mathcal{O}$. To actually recover $\mathbf{A}$ from $\mathbf{A}\mathbf{A}^T$ we propose Algorithms 4 and 5, appear in Appendix A, for the uniform and i.i.d. ensembles, respectively.

**Worst-case Model.** While the above generative models allow us to obtain elegant theoretical results, they may be too idealistic for real-world clustering applications. To this end, we also consider a general clustering model where each element in $\mathcal{N}$ can be belong to at most $\Delta \leq k$ clusters. Note that here each element may belong to *different* number of clusters. We show that under certain geometric conditions on the clusters, recovery is possible using a simple, efficient, and parameter free algorithm.

**Algorithm 1** Findmembership The algorithm for extracting the clustering matrix via queries to oracle.

---

**Require:** Number of elements: $N$, number of clusters $k$, oracle responses $\mathcal{O}_{\text{direct}}(i, j)$ for query $(i, j) \in \Omega$, where $i, j \in [N]$.
 1: Choose a set $\mathcal{S}$ of elements drawn uniformly at random from $\mathcal{N}$, and perform all pairwise queries corresponding to these $|\mathcal{S}|$ elements.
 2: Extract the membership of all the $|\mathcal{S}|$ elements and find representatives $\mathcal{T}$ for the $k$ clusters.
 3: Query each of the remaining $n - |\mathcal{S}|$ elements with all elements present in $\mathcal{T}$.
 4: Return the clusters.

---

**Algorithm 2** FindSimilarity The algorithm for extracting the similarity matrix $\mathbf{A}\mathbf{A}^T$ via queries to oracle.

---

**Require:** Number of elements: $N$, number of clusters $k$, oracle responses $\mathcal{O}_{\text{direct}}(i, j)$ for query $(i, j) \in \Omega$, where $i, j \in [N]$.
 1: Choose a set $\mathcal{S}$ of elements drawn uniformly at random from $\mathcal{N}$, and perform all pairwise queries corresponding to these $|\mathcal{S}|$ elements.
 2: Extract a valid membership of all the $|\mathcal{S}|$ elements by rank factorization of $\mathbf{A}_{\mathcal{S}}\mathbf{A}_{\mathcal{S}}^T$. Then, find a set $\mathcal{T} \subseteq \mathcal{S}$ that forms a basis of $\mathbb{R}^k$.
 3: Query each of the remaining $n - |\mathcal{S}|$ elements with all elements present in $\mathcal{T}$. Subsequently solve for the membership vector of the unknown element.
 4: Return the similarity matrix $\mathbf{A}\mathbf{A}^T$.

---

## 3 Algorithms and Their Performance Guarantees

In this section, we present our main theoretical results. Specifically, in Subsections 3.1 and 3.2 our main results about generative models are given. Subsection 3.3 is devoted to the worst-case model. Due to space limitations, a summary table showing our lower and upper bounds on the sample complexities, theoretical results concerning dithered oracle responses, information-theoretic lower bounds, as well as most of our pseudo-codes algorithms, appear in the appendices.

### 3.1 Direct Responses

As a warm up we start with the case of disjoint clusters. In this case, no statistical generative assumption on $\mathbf{A}$ is needed. The simple algorithm (Algorithm 1) for this serves as a building block for the other more complicated scenarios considered in this paper.

**Proposition 1.** *There exists a poly-time algorithm, i.e. Algorithm 1, which with probability at least $1 - n^{-\varepsilon}$ recovers exactly the set of clusters $\mathcal{N}_1, \ldots, \mathcal{N}_k$, $\mathcal{N}_i \cap \mathcal{N}_j = \emptyset$, for $i \neq j$, using $|\Omega| \geq k \cdot (n - m) + \binom{m}{2}$ queries, $m = (n/n_{\min}) \log(kn^\varepsilon)$, where $\varepsilon > 0$ and $n_{\min}$ is the size of the smallest cluster.*

*Proof Outline:* Pick $m$ elements uniformly at random from $\mathcal{N}$, and perform all $\binom{m}{2}$ pairwise queries among these $m$ elements. It can be shown that if $m \geq (n/n_{\min}) \log(kn^\varepsilon)$, then with probability $1 - n^{-\varepsilon}$, among these $m$ elements there will exist at least one element (representative) from each cluster. Finally, for the remaining $(n - m)$ items, we perform at most $k$ queries to decide which cluster they belong to. $\qquad\square$

From Prop. 1, when $n_{\min} = \Omega(n/k)$, the number of queries needed are $\Omega(kn)$. This result should be contrasted with standard matrix completion results with uniform sampling, which state that $O(kn \log n)$ queries are needed [8]. Next, we consider the overlapping case, where $\mathcal{N}_i \cap \mathcal{N}_j \neq \emptyset$. In this case the similarity matrix $\mathbf{A}\mathbf{A}^T$ is not binary anymore. For a set $\mathcal{S} \subseteq [n]$, with $m = |\mathcal{S}|$, we let $\mathbf{A}_{\mathcal{S}}$ be the $m \times k$ projection matrix formed by the rows of $\mathbf{A}$ that correspond to the indices in $\mathcal{S}$. We have the following result.

**Theorem 1.** *There exists a polynomial-time algorithm, given in Algorithm 2, which with probability at least $1 - \text{poly}(n^{-1})$ recovers exactly the set of (overlapping) clusters $\mathcal{N}_1, \ldots, \mathcal{N}_k$, using $|\Omega| \geq \binom{|\mathcal{S}|}{2} + k \cdot (n - |\mathcal{S}|)$ queries, where $|\mathcal{S}| > \mathsf{S}_{\text{uniform}} \triangleq \frac{\binom{k}{\Delta}}{\binom{k-\Delta}{\Delta-1}}[1 + c_1 \log k + c_2 \log n]$, for the uniform*

*ensemble, and* $|\mathcal{S}| > \mathsf{S}_{\text{i.i.d.}} \triangleq k - 1 - \frac{\log k + c_3 \log n}{\log \max(p, 1-p)}$, *for the i.i.d. ensemble, with* $c_1, c_2, c_3 > 0$ *arbitrary positive numbers.*

Let us explain the main idea behind Theorem 1. It is evident from Algorithm 2 that as long as we get a valid subset of elements $\mathcal{T} \subseteq \mathcal{S}$ whose membership vectors form a basis of $\mathbb{R}^k$, then querying a particular element $i \in \mathcal{N}$ with all elements in $\mathcal{T}$ gives $k$ linearly independent equations in $k$ variables that denote the membership of $i$th element to the different clusters. Subsequently, we can solve this system of equations uniquely to obtain the membership vector of $i$th element. Hence, if we choose $|\mathcal{S}|$ such that there exists a valid subset of $\mathcal{S}$ forming a basis of $\mathbb{R}^k$ with high probability, then we will be done and the sample complexity will be $\binom{|\mathcal{S}|}{2} + k(n - |\mathcal{S}|)$. Lemmas 5 and 6 (see, Appendix B), respectively, show that if $|\mathcal{S}| > \mathsf{S}_{\text{uniform}}$ for the uniform ensemble, and $|\mathcal{S}| > \mathsf{S}_{\text{i.i.d.}}$ for the i.i.d. ensemble, then the above property holds.

**Remark 2.** *Note that in the second step of Algorithm 2 we perform a rank factorization of the matrix* $\mathbf{A}_{\mathcal{S}} \mathbf{A}_{\mathcal{S}}^T$. *However, this factorization is not guaranteed to be unique, and accordingly, the resultant rank factorized matrix might be wrong. However, we show in the supplementary material, that even if this is the case, Algorithm 2 will nevertheless recover the true similarity matrix.*

### 3.2 Quantized Noisy Responses

We next move to the case where the oracle responses are quantized and noisy, namely, when queried with $(i,j)$, the oracle output is $\mathcal{O}_{\text{quantized}}(i,j) = \mathcal{Q}\left(\mathbf{A}_i^T \mathbf{A}_j\right) \oplus W_{i,j}$, where $W_{i,j} \sim \text{Bernoulli}(q)$. We start with the uniform ensemble, for which we have the following result.

**Theorem 2.** *Assume that* $\mathbf{A}$ *was generated according to the uniform ensemble, with* $k \geq 3\Delta$. *Then, there exists a polynomial-time algorithm, given in Algorithm 6, which with probability* $1 - n^{-\epsilon}$, *recovers the similarity matrix* $\mathbf{A}\mathbf{A}^T$, *using* $|\Omega| \geq \binom{|\mathcal{S}|}{2} + |\mathcal{S}| \cdot (n - |\mathcal{S}|)$ *queries, where for any* $\varepsilon > 0$,

$$|\mathcal{S}| > 2(1 - 2q)^{-4} \binom{k}{\Delta}^2 \left[ \binom{k - 2\Delta + 1}{\Delta} - \binom{k - 2\Delta}{\Delta} \right]^{-2} \log(2n^{2+\varepsilon}). \tag{2}$$

The main idea behind Algorithm 6 is the following: we first choose a random subset $\mathcal{S} \subseteq \mathcal{N}$ of elements, such that (2) holds, and perform all pairwise queries among these elements. Using the resultant queries we infer the unquantized inner products of $\mathbf{A}_i^T \mathbf{A}_j$, for any $(i,j) \in \mathcal{S}$. To this end, we count the number of elements which are similar to both the profile of elements $i$ and $j$ (see, the definition of $T_{ij}$ in Algorithm 6). Intuitively, it makes sense that the more similar the two elements $i$ and $j$ themselves are, the more the number of elements should be which are similar to both of them. We show that the condition in (2) suffices to make the count highly concentrated around its mean, and accordingly, outputs the true value of $\mathbf{A}_i^T \mathbf{A}_j$. Finally, the remaining $(n - |\mathcal{S}|)$ elements are queried with the elements in $\mathcal{S}$, and then we apply the above inferring procedure once again. We emphasize here that the exponential dependency of the upper bounds on $\Delta$ is inherent, as the information-theoretic lower bounds in Appendix I suggest.

It turns out that the above idea is capable to handle the other scenarios considered in this paper, albeit with certain technical modifications. Indeed, for the i.i.d. ensemble, we need an additional step before we can use the idea mentioned above. This is mainly because of the fact that analyzing the aforementioned count statistic requires the knowledge of support size of $\mathbf{A}_i$ and $\mathbf{A}_j$ (which is fixed in the uniform ensemble). An easy way around this problem is to infer first the $\ell_0$-norm of every element by counting the number of other elements that are similar. As before, under certain conditions, this count behaves differently for different values of the actual $\ell_0$-norm value and therefore we can infer the correct value. Once this step is done, everything else falls into place. Due to space limitation we relegate the pseudo-algorithm for the i.i.d. setting to the appendices. We have the following result.

**Theorem 3.** *Assume that* $\mathbf{A}$ *was generated according to the i.i.d. ensemble. Then, there exists a polynomial-time algorithm, given in Algorithm 9, which with probability* $1 - n^{-\epsilon}$, *recovers the similarity matrix* $\mathbf{A}\mathbf{A}^T$, *using* $|\Omega| \geq \binom{|\mathcal{S}|}{2} + |\mathcal{S}| \cdot (n - |\mathcal{S}|)$ *queries, where for any* $\varepsilon > 0$,

$$|\mathcal{S}| > 2p^{-2}(1 - 2q)^{-4}(1 - p)^{2-2k} \log(2n^{2+\varepsilon}). \tag{3}$$

In practice, the value of the noise parameter $q$ might be unknown to the learner. In this case, we will not know the expected values of the triangle counts under the different hypotheses a-priori, and thus

our previous algorithms cannot be used directly. Fortunately, however, it turns out that with a simple modification, our algorithms can be used also when $q$ is unknown. We have the following result stated for the uniform ensemble. A similar result can be obtained also for the i.i.d. ensemble.

**Theorem 4.** *Assume that $\mathbf{A}$ was generated according to the uniform ensemble with $k \geq 3\Delta$, and $n > 10\binom{k}{\Delta} \log n$. Then, there exists a polynomial-time algorithm, given in Algorithm 14, independent of the noise parameter $q$, which with probability $1 - n^{-\epsilon}$, recovers the similarity matrix $\mathbf{A}\mathbf{A}^T$, using $|\Omega| \geq \binom{|\mathcal{S}|}{2} + |\mathcal{S}| \cdot (n - |\mathcal{S}|)$ queries, where for any $\varepsilon > 0$,*

$$|\mathcal{S}| > 18(1-2q)^{-4} \binom{k}{\Delta}^2 \left[ \binom{k-2\Delta+1}{\Delta} - \binom{k-2\Delta}{\Delta} \right]^{-2} \log(2n^{2+\varepsilon}). \qquad (4)$$

Comparing Theorems 2 and 4, we notice that the query complexity grows by a multiplicative constant factor only. Note that the additional technical condition $n > 10\binom{k}{\Delta} \log n$ is rather weak, and naturally satisfied, for example, in the regime $k = O(\log n)$. Finally, note that since we deal with quantized responses without any continuous dithering, matrix completion results cannot be used. In fact, without dithering matrix completion algorithms will fail on quantized data [12], as they do not exploit the discrete structure of the data, which is the main source for the success of our algorithms.

### 3.3 Beyond Generative Models: Arbitrary Worst-Case Instances

In this subsection, we consider the worst-case model, where we do not impose any statistical assumptions, and assume that each element belong to at most $\Delta$ clusters. We focus on noiseless quantized oracle responses, but also discuss the direct responses scenario in Section 4. For this case, we propose Algorithm 17. We have the following result.

**Theorem 5.** *Let $\mathcal{N}_i$ be the set of elements which belong to the $i$'th cluster. If, for every cluster $i \in [k]$, we have $|\mathcal{N}_i \setminus \{\bigcup_{j:j\neq i} \mathcal{N}_j\}| > \alpha \cdot n$, for some $\alpha > 0$, then by using Algorithm 3, $\binom{|S|}{2} + |S| \cdot (n - |S|)$ queries are sufficient to recover the clusters, where $\alpha \cdot |S| = \log k + \log n$.*

As mentioned above, Algorithm 3 is parameter free, do not require the knowledge of $\Delta$, and efficient. For the special case of $\Delta = 2$, we show in Appendix J (see, Theorem 9) that the same result holds under less restrictive conditions than those in Theorem 5. In fact, in Appendix K we conjecture that Theorem 5 holds true under a similar assumption as in Theorem 9. Depending on the dataset, the scaling of $\alpha$ in Theorem 5 w.r.t. $(\Delta, k, n)$ may vary widely. For example, in the non-overlapping case, $\alpha = k_{\min}/n \leq 1/k$, where $k_{\min}$ is the size of the smallest cluster, which implies that the query complexity in the best scenario is $O(nk \log n)$, which is consistent with our results in the previous section. In the worst-case, a positive $\alpha$ could be as small as $1/n$ (unreasonable in real-world datasets), which implies a query complexity of $O(n^2)$. This is much higher than our average case results, as expected. More generally, note that $\alpha$ decreases as a function of $\Delta$, which implies that the query complexity increases with $\Delta$. For example, consider the example of 3 equally-sized clusters $A$, $B$ and $C$. Suppose $\Delta = 1$ and in that case $|A \setminus B \cup C| = |A| = n/3$, implying that $\alpha = 1/3$. Now suppose that $\Delta = 2$. In this case $A \cap B$ and $A \cap C$ are non-empty and therefore $|A \setminus B \cup C| = |A| - |A \cap B| - |A \cap C| < n/3$, namely, $\alpha$ is less than $1/3$.

---

**Algorithm 3** `Worst-case quantized responses`

---

**Require:** $N, k$, and oracle responses $\mathcal{O}_{\mathsf{quantized}}(i, j)$ for every query $(i, j) \in \Omega$.
 1: Choose a set $\mathcal{S}$ of elements drawn uniformly at random from $[N]$, and perform all pairwise queries corresponding to these $|\mathcal{S}|$ elements.
 2: Construct a graph $\mathcal{G} = (\mathcal{V}, \mathcal{E})$ where the vertices are the $|\mathcal{S}|$ sampled elements. There exist an edge between elements $(i, j)$ only if they are determined to be similar by the oracle.
 3: Construct the maximal cliques of the graph $\mathcal{G}$ such that all edges in $\mathcal{E}$ are covered. Each maximal clique forms a cluster.
 4: Query each of the remaining $n - |\mathcal{S}|$ elements with all elements present in $\mathcal{S}$. For each cluster, if an element is similar with all the elements in that particular cluster, then assign the element to that cluster. Return the obtained clusters.

---

Table 1: Sample complexities for $k = O(\log n)$ and $\Delta \ll k$

| Oracle Type | Lower-Bound | Upper-Bound |
|---|---|---|
| Direct responses (disjoint) | $O(nk)$ | $\Omega(nk)$ |
| Direct responses (overlapping) | $O(nk)$ | $\Omega(nk)$ |
| Quantized responses | $O(n \cdot \text{polylog } n)$ | $\Omega(n \cdot \text{polylog } n)$ |
| Quantized responses (worst-case, $\alpha = n^{-c}$) | NA | $\Omega(n^{1+c} \cdot \text{polylog } n)$ |

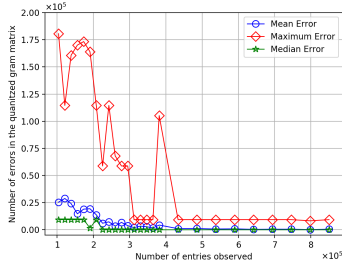

(a) Mean, median and maximum errors for $\Delta = 2$.

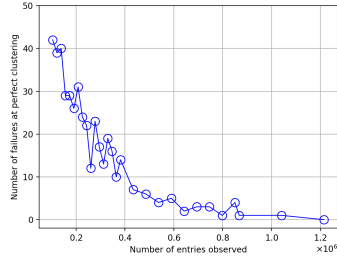

(b) Number of failures for $\Delta = 2$.

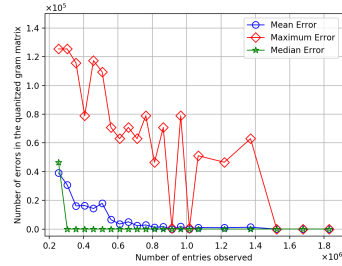

(c) Mean, median and maximum errors for $\Delta = 3$.

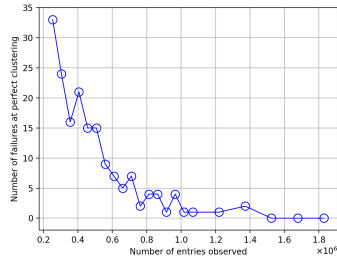

(d) Number of failures for $\Delta = 3$.

Figure 1: Results of our techniques on MovieLens dataset.

## 3.4 Summary Table

Table 1 summarizes the scaling of our lower and upper bounds on the sample complexities for each of the different oracle types considered in this paper. In the table, we opted to focus on the regime where $k = O(\log n)$ and $\Delta \ll k$, as we found it to be the most interesting one. We also assume that the noise parameter $q$ is fixed. Note, however, that our theoretical results are general and apply for any scaling of $k$ and $\Delta$. Also, since the scalings of the sample complexities associated with the uniform and the i.i.d. ensembles, as well as when the noise parameter $q$ is unknown, are similar, we choose to combine them together. We also present the worst-case scenario in Theorem 5, assuming that $\alpha = n^{-c}$, for some $c \in (0, 1)$. For simplicity of presentation, we do not explicitly present the scaling of the lower and upper bound on polylog factors. For the regime above, we can see that the scaling of the upper and lower bounds w.r.t. $n$ is the same up to constants for direct responses. For quantized responses there is a polylog factor difference between the obtain upper and lower bounds.

## 4 Experimental Results

We provide only a summary of our experimental results here, deferring synthetic data results to Appendix L. We focus on real-world data from the popular Movielens dataset (`https://grouplens.org/datasets/movielens/`) for our experiments. The dataset we used describes 5-star rating and free-text tagging activity from Movielens (`http://movielens.org`), a movie recommendation service. It contains 100836 ratings and 3683 tag applications across 9742 movies.

**Quantized Query Responses.** In order to establish our results we chose the following categories `Mystery`, `Drama`, `Sci-Fi`, `Horror`, and `Crime`, and first selected only those movies that belong to at most two categories and at least one category (i.e., $\Delta = 2$). The total number of such movies were $3470$ and in accordance to the statement in Theorem 9, we have $\alpha = 0.0152$ (there are 53 movies that belong to `Mystery` but does not belong to `Sci-fi` and `Horror`). The total number of possible queries is about $1.2 \times 10^7$ and the number of queries that are sufficient theoretically is $2948935$ (theoretical value of $|\mathcal{S}|$ is 245). We ran Algorithm 3 (running Algorithm 17 requires parsing all possible clique covers which is computationally hard) with different values of $|\mathcal{S}|$ (number of movies randomly chosen in the first step) and since the movies are sampled randomly, we ran 50 trials for each value of $|\mathcal{S}|$. Finally, after the final clustering is provided by the algorithm, we calculated the gram matrix from the resulting clustering and compared it with the gram matrix of the ground truth clustering. Figure 1a shows the mean, median, and maximum error as a function of the total number of queries accrued by Algorithm 3. Here, the error refers to the total number of different entries in the estimated and true gram matrices. We can observe that the mean error almost reaches zero around $3 \times 10^5$ queries (about $2.5\%$ of total). Figure 1b presents the total number of failures in perfect clustering (trials when error is larger than 1) among the 50 trials for each value of $|\mathcal{S}|$ we have chosen. We obtain perfect clustering in all the 50 trials first using $1.2 \times 10^6$ queries ($\approx 10\%$ of total). Note that since Theorem 5 gives a sufficient condition on $T$ only, in practice we can take smaller values for $T$ and still guarantee recovery. Of course, the smaller the size of $\mathcal{S}$ is, the sample and computational complexities are smaller as well. We repeated the experiment with the same set of categories as in the previous one but this time, we included movies that belonged to at most three clusters at the same time i.e ($\Delta = 3$). The total number of such movies is $5082$ and therefore the total number of possible queries is about $2.59 \times 10^7$. Again, we conducted 50 trials for each chosen value of $|\mathcal{S}|$ and as before, we plotted the mean, median and maximum error in Figure 1c and the number of failures in perfect clustering in Figure 1d for this setting. Notice that the mean error drops almost to zero at about $6 \times 10^5$ queries ($2.32\%$ of total) and perfect clustering over all 50 trials is achieved at $1.5 \times 10^6$ queries ($5.8\%$ of total). We would like to point out over here that Algorithm 3 is *parameter free*, and provides a non-trivial solution even when the number of queries are far below the theoretical threshold limit. It turns out that the experimental threshold is better than the theoretical threshold on queries for perfect clustering. Moreover, Algorithm 3 is efficient in partial clustering as well since the error drops very fast as the number of queries is increased.

**Unquantized Query Responses.** In this experiment, we used the Movielens dataset again and chose 5 classes `Mystery`, `Drama`, `IMAX`, `Sci-Fi`, and `Horror`, and selected those elements who belonged to at most two categories and at least one category (i.e., $\Delta = 2$). The total number of such movies are $3270$, and for each query involving two movies, we obtain back the unquantized similarity (total number of categories they both belong to). We follow Algorithm `FindSimilarity` very closely but with a small modification. Indeed, note that Algorithm 2 is designed so that the guarantees hold under a specific stochastic assumption. More concisely, the necessary size of $\mathcal{S}$ is not defined for arbitrary real-world datasets. Note, however, that the main objective in the first part of the algorithm is to select a number of elements so that the gram matrix is of full rank. Therefore, for a real-world dataset, instead of sampling a fixed number of movies a-priori, we randomly select $k = 5$ movies and make all pairwise queries restricted to those 5 movies. Then, we check if the $5 \times 5$ gram matrix (with the $(i, j)$ entry being the unquantized similarity between the $i$th and $j$th movies) is of rank 5 and if yes, then we will use that matrix for further calculations. If not, we sample again until we succeed. We then proceed to factorize the obtained $5 \times 5$ gram matrix into the form of $\mathbf{BB}^T$, where $\mathbf{B}$ is binary. Finally, we query every movie with the 5 movies already sampled (and clustered). This provides us with five linearly independent equations in five variables (each corresponding to whether the movie belongs to a particular cluster). Solving the equations for each movie, we finally obtain the categories each movie belongs to. Hence, the number of queries is at most

$$\text{Number of trials to obtain a rank 5 matrix} \times 25 + 3270 \times 5.$$

Since the algorithm is randomized, we simulated this 50 times and we found that the Mean query complexity is $232126$ (with a standard deviation of $269315.36$) which is only $4.34\%$ of the total number of possible queries.

**Acknowledgements:** This research is supported in part by NSF Grants CCF 1642658, 1642550, 1618512, and 1909046.

## Footnotes

[1]Given $\mathbf{A}\mathbf{A}^T$, it is only possible to recover the clustering matrix $\mathbf{A}$ up to a permutation of columns, see Section 2.

[2]The case where different items may belong to different numbers of clusters can be handled using the same techniques developed in this paper.

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
