[Supplementary Material · Supp_Material.pdf]

# Supplementary Material: Same-Cluster Querying for Overlapping Clusters

## A  Uniqueness of Factorization

In this section, we prove that under certain conditions, the clustering matrix $\mathbf{A}$ can be uniquely (up to a permutation of columns) recovered from $\mathbf{A}\mathbf{A}^T$, for both the uniform and i.i.d. ensemble. Specifically, we have the following two results, proved in the sequel.

**Lemma 3.** *[Uniform Ensemble Uniqueness] Let $k \geq 2\Delta - 2$, $\Delta > 2$, and $n > c \cdot \binom{k}{\Delta} \log \binom{k}{\Delta}$, for some $c > 0$. Consider two $n \times k$ binary matrices $\mathbf{A}$ and $\mathbf{B}$, drawn from the uniform ensemble, and assume that $\mathbf{A}\mathbf{A}^T = \mathbf{B}\mathbf{B}^T$. Then, $\mathbf{B}$ is a column-permuted version of $\mathbf{A}$, namely, $\mathbf{B} = \mathbf{A}\mathbf{P}$, where $\mathbf{P}$ is a permutation matrix, with overwhelming probability.*

**Lemma 4.** *[i.i.d. Ensemble Uniqueness] Let $\mathbf{A}$ and $\mathbf{B}$ be two $n \times k$ binary matrices, drawn from the i.i.d. ensemble with parameter $p$. Assume that $\mathbf{A}\mathbf{A}^T = \mathbf{B}\mathbf{B}^T$. If $n > \frac{c\log n + \log k}{1 - p\cdot(1-p)^k}$ for some $c > 0$, then $\mathbf{B}$ is a column-permuted version of $\mathbf{A}$, namely, $\mathbf{B} = \mathbf{A}\mathbf{P}$, where $\mathbf{P}$ is a permutation matrix, with overwhelming probability.*

To actually recover $\mathbf{A}$ from $\mathbf{A}\mathbf{A}^T$ we propose Algorithms 4 and 5, for the uniform and i.i.d. ensembles, respectively. It can be shown that the worst-case computational complexities of these algorithms are $O\left(nk^3 \left(\frac{k}{\Delta}\right)^k\right)$ and $O\left(\binom{n}{k}\right)$, respectively. This means that the when $k$ is fixed, the computational complexities are polynomial in $n$, while if $k$ grows with $n$, e.g., $k = O(\log n)$, then the computational complexities are quasi-polynomial in $n$

### A.1  Proof of Lemma 3

First, note that if $n > c\binom{k}{\Delta} \log \binom{k}{\Delta}$ for some constant $c > 1$, then all vectors in $T_k(\Delta)$ will be present in $\mathbf{A}$, with high probability, by a simple coupon collector argument.

*Idea of the solution for $\Delta > 2$:* We can think of the mapping from $\mathbf{A}_i$ (ith row of $\mathbf{A}$) to $\mathbf{B}_i$ (ith row of $\mathbf{B}$) as a permutation $\sigma_i$ of the columns i.e.

$$\mathbf{B}_i = \sigma_i(\mathbf{A}_i).$$

Notice that there can be multiple permutations $\sigma_i$ that can explain the mapping $\mathbf{A}_i \rightarrow \mathbf{B}_i$. A permutation $\sigma$ can be described a series of swaps $\{(i_k, j_k)\}$ which implies that at the $kth$ step, the element in the $i_k^{th}$ position is swapped with the element in the $i_k^{th}$ position. The composition of permutation $\sigma_1$ with $\sigma_2$ implies implementing the swaps corresponding to $\sigma_2$ after the swaps corresponding to $\sigma_1$. Now, if there exists a permutation $\sigma$ which is the same for all rows $i$, then definitely $\mathbf{B}$ can be constructed by a permutation of the columns of $\mathbf{A}$. In fact if $\sigma_i$ is the same for at least $k$ linearly independent rows of $\mathbf{A}$, then $\sigma_i$ is same for all rows of $\mathbf{A}$ because it uniquely defines the rotation matrix $\mathbf{R}$. Hence if one can construct a set of at least $k$ $\Delta$-sparse vectors which are

---

**Algorithm 4** Factorization1 Algorithm for recovering $\mathbf{A}$ from $\mathbf{A}\mathbf{A}^T$, when rows of $\mathbf{A}$ belong to $T_k(\Delta)$

---

**Require:** Similarity matrix $\mathbf{A}\mathbf{A}^T$.
 1: Find a full rank $k \times k$ binary submatrix $\mathbf{T}$ of $\mathbf{A}\mathbf{A}^T$.
 2: Find the set of matrices $\mathcal{Q} \in \{0,1\}^{k \times k}$ such that for any $\mathbf{Q} \in \mathcal{Q}$, $\mathbf{Q}\mathbf{Q}^T = \mathbf{T}$, and $\|\mathbf{Q}_i\|_0 = \Delta$, $\forall i$.
 3: **for** $\mathbf{Q} \in \mathcal{Q}$ **do**
 4:    For the remaining $n - k$ elements, consider the inferred values with the row indices in $\mathbf{T}$. This creates a system of $k$ linear equations which can be solved for the membership vector of that element.
 5:    **if** All membership vectors belong to $T_k(\Delta)$ **then**
 6:        Exit the outer FOR loop
 7: Return the matrix $\mathbf{A}$.

---

**Algorithm 5** `Factorization2` Algorithm for recovering $\mathbf{A}$ of $\mathbf{A}\mathbf{A}^T$ when elements of $\mathbf{A}$ are i.i.d Bernoulli($p$) random variables

---

**Require:** Similarity matrix $\mathbf{A}\mathbf{A}^T$.
 1: Find a $k \times k$ submatrix $\mathbf{T}$ of $\mathbf{A}\mathbf{A}^T$ formed by all $k$-dimensional binary unit vectors.
 2: Choose a permutation matrix $\mathbf{P}$ which forms the membership vector for the indices of the rows in $\mathbf{T}$.
 3: For each remaining $n - k$ elements, consider the inferred values with the row indices in $\mathbf{T}$. This creates a system of $k$ linear equations which can be solved for the membership vector of that element.
 4: Return the matrix $\mathbf{A}$.

---

linearly independent and their gram matrix will have a unique factorization, then we are done (recall that those vectors will be in $\mathbf{A}$ since $\mathbf{A}$ contains all vectors from $T_k(\Delta)$).

*Construction:* Consider the following matrices $C_1 \equiv [D\ I]$ and $C_2 \equiv [I\ D]$ of dimension $k - \Delta + 1 \times k$. Here $D$ is a matrix of dimension $k - \Delta + 1 \times \Delta - 1$ and all the entries of $D$ is 1 and $I$ is the identity matrix. The matrix that we will use for our construction is the following:

$$Q \equiv \begin{bmatrix} C_1 \\ C_2 \end{bmatrix}.$$

The number of rows in $Q$ is at least $k$ since we know that $k \geq 2\Delta - 2$. Notice that the only possible solution for the gram matrix of $C_1$ (and $C_2$) is of the form $[P_{11}\ D_{11}\ D_{12}\ \ldots\ P_{1r}]$ where $[P_{11}\ P_{12}\ P_{13}\ldots\ P_{1r}]$ forms a permutation matrix and $[D_{11}\ D_{12}\ \ldots\ D_{1r'}] = D$ (This is true only for $\Delta > 2$. In order to see why, consider the rows of $C_1$ and think of a game between these rows where the first row makes a swap in its columns and then all the rows tries to make swaps so that the inner products remain preserved. Suppose the first row makes a swap in the columns present in the span of $I$ in which case the other rows have to make the same swap. Now suppose the first row makes a swap in the columns present in the span of $D$ which is equivalent to no swap at all. Lastly, suppose the first row makes a swap in the two columns in which one belongs to $I$ and the other belongs to $D$. Again it can be checked that the other rows has to make the same swap to preserve the inner product. This implies that there exists a set of permutations $\Sigma_1$ ($\Sigma_2$) that explains the mapping of all rows in $C_1$ ($C_2$). If we can only show that there must exist a permutation $\sigma$ that belongs to both $\Sigma_1$ and $\Sigma_2$, then we are done. Suppose the new solution is

$$\hat{Q} \equiv \begin{bmatrix} \hat{Q}_1 \\ \hat{Q}_2 \end{bmatrix} \equiv \begin{bmatrix} P_{11}\ D_{11}\ D_{12}\ \ldots\ P_{1r} \\ P_{21}\ D_{21}\ D_{22}\ \ldots\ P_{2s} \end{bmatrix}.$$

The set of columns containing $D_{11}, D_{12}, \ldots, D_{1r'}$ and the columns containing $D_{21}, D_{22}, \ldots, D_{2s'}$ are disjoint otherwise there must exist two rows in $Q$ whose inner product violates its original value. 1) For $k \geq 2\Delta$, there exists two rows in $C_1$ and $C_2$ whose inner product is 0 which cannot be the case if the columns are not disjoint. 2) For $k = 2\Delta - 1$, there exists a row in $C_2$ whose inner product is 1 with all the rows in $C_1$. Again this cannot be the case if the columns are not disjoint. 3) For $k = 2\Delta - 2$, the inner product between any row in $C_1$ and $C_2$ is 2 which is not possible if the columns are not disjoint). Therefore if we fix $\hat{Q}_1$, then the inner product of the any row of $\hat{Q}_2$ with all the rows in $\hat{Q}_1$ exactly specifies the position of the 1's in all the columns except the columns spanned by $D_{11}, D_{12}, \ldots, D_{1r'}$. Hence a permutation in $\Sigma_1$ exactly specifies all the swaps in $\Sigma_2$ except those swaps restricted to the columns spanned by $D_{11}, D_{12}, \ldots, D_{1r'}$ (again can be verified by a simple case study of the three cases: 1) $k = 2\Delta - 2$, 2) $k = 2\Delta - 1$, and 3) $k > 2\Delta - 1$. However, if $\Sigma_1$ contains a particular permutation, then a composition of that permutation with other permutations that only contains swaps restricted to the columns in $D_{11}, D_{12}, \ldots, D_{1r'}$ does not change the mapping from $C_1 \to \hat{Q}_1$ and therefore $\Sigma_1$ contains these permutations as well. Hence there exist a single permutation $\sigma$ in $\Sigma_1$ and $\Sigma_2$ that explains the mapping from $Q \to \hat{Q}$. Hence the proof is complete.

### A.2   Proof of Lemma 4

Let $\mathbf{e}_i \in \{0, 1\}^k$ be the $k$-dimensional binary unit vector, namely, $\mathbf{e}_i$ is all zero except at its $i$th position. Suppose that all the $\{\mathbf{e}_i\}_{i=1}^k$ vectors are present in the matrix $\mathbf{A}$, and let us denote the

sub-matrix formed by these unit vectors by $\mathbf{Q}$. Thus, $\mathbf{Q}$ is a $k \times k$ matrix with rank $k$. It is easy to see that for any $k \times k$ binary matrix $\mathbf{R}$ such that $\mathbf{R}\mathbf{R}^T = \mathbf{Q}\mathbf{Q}^T$, $\mathbf{R}$ can be constructed by a permutation of the columns of $\mathbf{Q}$. Hence if the event "$\mathbf{Q}$ is a sub-matrix of $\mathbf{A}$" is true, then for any matrix $\mathbf{B}$ such that $\mathbf{B}\mathbf{B}^T = \mathbf{A}\mathbf{A}^T$, $\mathbf{B}$ can be constructed by a permutation of the columns of $\mathbf{A}$. Let $\mathsf{E}_i$ denote the event that the vector $\mathbf{e}_i$ is not present in $\mathbf{A}$. Therefore,

$$
\begin{aligned}
\mathbb{P}\left(\mathbf{Q} \text{ is sub-matrix of } \mathbf{A}\right) &= 1 - \mathbb{P}\left(\bigcup_{i=1}^{k} \mathsf{E}_i\right) \\
&\geq 1 - k \cdot \mathbb{P}(\mathsf{E}_1) \\
&= 1 - k \cdot \left[1 - p \cdot (1-p)^{k-1}\right]^n \\
&\geq 1 - k \cdot e^{-c \log n - \log k} \\
&= 1 - \frac{1}{n^c}
\end{aligned}
\tag{5}
$$

where the last inequality follows by substituting the condition on $n$ in the theorem statement.

## B    The Rank of Random Matrices

In this section, we state two important lemmas concerning the rank of the clustering matrix under both ensembles. For a set $\mathcal{S} \subseteq [n]$, with $m = |\mathcal{S}|$, we let $\mathbf{A}_{\mathcal{S}}$ be the $m \times k$ projection matrix formed by the rows of $\mathbf{A}$ that correspond to the indices in $\mathcal{S}$.

**Lemma 5.** *[Rank of Uniformly Drawn Matrix] Let $\mathbf{A}$ be a random matrix drawn uniformly from $T_k(\Delta)$. Also, let $\mathcal{S}$ be a set of size $m > k$, drawn uniformly at random from $[n]$. Then, if $m \geq \frac{\binom{k}{\Delta}}{\binom{k-\Delta}{\Delta-1}}[1 + c_1 \log k + c_2 \log n]$, for some $c_1 > 1$ and $c_2 > 0$, then*

$$
\mathbb{P}\left[\mathrm{rank}_{\mathbb{R}}(\mathbf{A}_{\mathcal{S}}) = k\right] \geq 1 - \frac{1}{n^{c_2} k^{c_1}}.
\tag{6}
$$

**Lemma 6.** *[Rank of i.i.d. Matrix] Let $\mathbf{A}$ be an i.i.d. matrix with $\mathsf{Bernoulli}(p)$ entries. Also, let $\mathcal{S}$ be a set of size $m > k$, drawn uniformly at random from $[n]$, and define $\alpha \triangleq \max(p, 1-p)$. Then,*

$$
\mathbb{P}\left[\mathrm{rank}_{\mathbb{R}}(\mathbf{A}_{\mathcal{S}}) = k\right] \geq 1 - \min\left(1, k \cdot \alpha^{m-k+1}\right).
\tag{7}
$$

The above results imply that by taking $m$ large enough (as a function of $k$, $\Delta$, and $n$) we can guarantee that a sub-matrix formed by a random subset of rows taken from $\mathbf{A}$ is of full rank with high probability. Specifically, for the i.i.d. ensemble, if

$$
|\mathcal{S}| > k - 1 - \frac{\log k + c \log n}{\log \max(p, 1-p)} \triangleq \mathsf{S}_{\mathsf{i.i.d.}},
\tag{8}
$$

for some $c > 0$, then, $\mathbb{P}\left[\mathrm{rank}_{\mathbb{R}}(\mathbf{A}_{\mathcal{S}}) = k\right] \geq 1 - n^{-c}$. Similarly, for the uniform ensemble, if

$$
|\mathcal{S}| > \frac{\binom{k}{\Delta}}{\binom{k-\Delta}{\Delta-1}}[1 + c_1 \log k + c_2 \log n] \triangleq \mathsf{S}_{\mathsf{uniform}},
\tag{9}
$$

for some $c_1 > 0$ and $c_2 > 0$, then, (6) holds.

### B.1    Proof of Lemma 5

Given $k, \Delta \in \mathbb{N}$, define the set

$$
T_k(\Delta) \triangleq \left\{\mathbf{c} \in \{0,1\}^k : w_H(\mathbf{c}) = \Delta\right\},
\tag{10}
$$

namely, the set of all $k$-length binary sequence with Hamming weight $\Delta$. Let $\mathbf{A}$ be an $n \times k$ matrix formed by drawing independently $n$ sequences from $T_k(\Delta)$ and putting those as rows of $\mathbf{A}$. Let $\mathcal{S}$ be a set of size $m > k$ drawn uniformly at random from $[1:n]$. Let $\mathbf{A}_{\mathcal{S}}$ be an $m \times k$ matrix with the

rows in $\mathbf{A}$ that correspond to the indices in $\mathcal{S}$. We would like to understand how large $m$ should be such that
$$\mathbb{P}\left[\text{rank}_{\mathbb{R}}(\mathbf{A}_{\mathcal{S}}) < k\right]$$
decays to zero as $\text{poly}(n^{-1})$. By symmetry, it is clear that

$$\mathbb{P}\left[\text{rank}_{\mathbb{R}}(\mathbf{A}_{\mathcal{S}}) < k\right] = \mathbb{P}\left[\text{rank}_{\mathbb{R}}(\mathbf{B}) < k\right] \tag{11}$$
$$\leq \mathbb{P}\left[\text{rank}_{\mathbb{F}_2}(\mathbf{B}) < k\right] \tag{12}$$

where $\mathbf{B}$ refers to a submatrix of $\mathbf{A}$ formed by taking, for example, the first $m$ rows, and the last inequality follows from the fact that for any filed $\mathbb{F}$, and any binary matrix $\mathbf{M}$ it holds $\text{rank}_{\mathbb{F}}(\mathbf{M}) \leq \text{rank}_{\mathbb{R}}(\mathbf{M})$.

We next analyze the probability term on the r.h.s. of the above inequality. To this end, we note that the event $\text{rank}_{\mathbb{F}_2}(\mathbf{B}) < k$ is in fact equivalent to the existence of a set $\mathcal{R}$ with $|\mathcal{R}| \leq k$ of column indices such that each row of $\mathbf{B}$ has an even number of 1's in $\mathcal{R}$. Indeed, if this is the case, then some columns will be linearly dependent and thus the rank must be smaller than $k$. Accordingly, given a set of column indices $\mathcal{R}$, let $\mathcal{E}_{\mathcal{R}}$ denote the event that each row of $\mathbf{B}$ has an even number of 1's in $\mathcal{R}$. Then, using the above observation and the union bound,

$$\mathbb{P}\left[\text{rank}_{\mathbb{F}_2}(\mathbf{B}) < k\right] = \mathbb{P}\left[\bigcup_{|\mathcal{R}| \leq k} \mathcal{E}_{\mathcal{R}}\right] \tag{13}$$

$$\leq \sum_{|\mathcal{R}|=1}^{k} \binom{k}{|\mathcal{R}|}\mathbb{P}(\mathcal{E}_{\mathcal{R}}). \tag{14}$$

It is left to understand the behavior of $\mathbb{P}(\mathcal{E}_{\mathcal{R}})$. The number of rows that have an odd number of non-zero elements in $\mathcal{R}$ is simply

$$N_{|\mathcal{R}|,\Delta,k} = \sum_{\ell:\text{ odd}}^{\Delta} \binom{|\mathcal{R}|}{\ell}\binom{k - |\mathcal{R}|}{\Delta - \ell} \tag{15}$$

following a simple counting argument. Accordingly,

$$\text{Pr}(\mathcal{E}_{\mathcal{R}}) = (1 - \alpha)^{N_{|\mathcal{R}|,\delta,k}} \tag{16}$$

where $\alpha \triangleq m \cdot [\binom{k}{\Delta}]^{-1}$. To get a simple upper bound on the probability of interest, we next lower bound $N_{|\mathcal{R}|,\Delta,k}$. It is clear that

$$N_{|\mathcal{R}|,\Delta,k} \geq \binom{|\mathcal{R}|}{1}\binom{k - |\mathcal{R}|}{\Delta - 1} \tag{17}$$

$$\geq |\mathcal{R}| \cdot \binom{k - \Delta}{\Delta - 1}. \tag{18}$$

Then,

$$\mathbb{P}\left[\text{rank}_{\mathbb{F}_2}(\mathbf{B}) < k\right] \leq \sum_{\ell=1}^{k} \binom{k}{\ell}(1 - \alpha)^{\ell \cdot \binom{k-\Delta}{\Delta-1}}$$

$$\leq \sum_{\ell=1}^{k} \left(\frac{ek}{\ell}\right)^{\ell} e^{\ell \log(1-\alpha) \cdot \binom{k-\Delta}{\Delta-1}}$$

$$\leq \sum_{\ell=1}^{k} e^{\ell\left[\log(ek) - \alpha \cdot \binom{k-\Delta}{\Delta-1}\right]}. \tag{19}$$

Now, taking $m = \frac{\binom{k}{\Delta}}{\binom{k-\Delta}{\Delta-1}}\left[\log(ek) + c_1 \log k + c_2 \log n\right]$, for some $c_1 > 1$ and $c_2 > 0$, we get that,

$$\mathbb{P}\left[\text{rank}_{\mathbb{F}_2}(\mathbf{B}) < k\right] \leq \frac{1}{n^{c_2}} \sum_{\ell=1}^{k} \frac{1}{k^{c_1\ell}} \leq \frac{1}{k^{c_1-1}n^{c_2}}. \tag{20}$$

## B.2 Proof of Lemma 6

Let $\mathbf{A}$ be an $n \times k$ i.i.d. matrix with each element distributed as $\mathsf{Bernoulli}(p)$, for some $0 < p < 1$. Let $\mathcal{S}$ be a set of size $m > k$ drawn uniformly at random from $[1 : n]$. Let $\mathbf{A}_{\mathcal{S}}$ be the $m \times k$ matrix formed by the rows of $\mathbf{A}$ that correspond to the indices in $\mathcal{S}$. We would like to understand how large $m$ should be such that

$$\mathbb{P}\left[\mathsf{rank}_{\mathbb{R}}(\mathbf{A}_{\mathcal{S}}) = k\right]$$

goes to one as $1 - \mathsf{poly}(n^{-1}, k^{-1})$. By symmetry, it is clear that

$$\mathbb{P}\left[\mathsf{rank}_{\mathbb{R}}(\mathbf{A}_{\mathcal{S}}) = k\right] = \mathbb{P}\left[\mathsf{rank}_{\mathbb{R}}(\mathbf{B}) = k\right] \tag{21}$$

where $\mathbf{B}$ refers to any submatrix of $\mathbf{A}$. Without loss of generality, let us take it to be formed by the first $m$ rows of $\mathbf{A}$. Also, we note that

$$\mathbb{P}\left[\mathsf{rank}_{\mathbb{R}}(\mathbf{B}) = k\right] = 1 - \mathbb{P}\left[\mathsf{rank}_{\mathbb{R}}(\mathbf{B}) < k\right] \tag{22}$$
$$\geq 1 - \mathbb{P}\left[\mathsf{rank}_{\mathbb{F}_2}(\mathbf{B}) < k\right] \tag{23}$$
$$= \mathbb{P}\left[\mathsf{rank}_{\mathbb{F}_2}(\mathbf{B}) = k\right] \tag{24}$$

where the inequality follows from the fact that for any filed $\mathbb{F}$, and any binary matrix $\mathbf{M}$ it holds $\mathsf{rank}_{\mathbb{F}}(\mathbf{M}) \leq \mathsf{rank}_{\mathbb{R}}(\mathbf{M})$. Therefore, it is suffice to lower bound $\mathbb{P}\left[\mathsf{rank}_{\mathbb{F}_2}(\mathbf{B}) = k\right]$.

Let $\mathcal{F}_i$ designate the event that the first $i$ columns of $\mathbf{B}$, denote by $\mathbf{B}_1, \ldots, \mathbf{B}_i$, are linearly independent. Then, it is clear that

$$\mathbb{P}\left[\mathcal{F}_{i+1}\right] = \mathbb{P}\left[\mathcal{F}_{i+1}|\mathcal{F}_i\right]\mathbb{P}\left[\mathcal{F}_i\right] + \mathbb{P}\left[\mathcal{F}_{i+1}|\mathcal{F}_i^c\right]\mathbb{P}\left[\mathcal{F}_i^c\right]$$
$$= \mathbb{P}\left[\mathcal{F}_{i+1}|\mathcal{F}_i\right]\mathbb{P}\left[\mathcal{F}_i\right] \tag{25}$$

where the second equality is because conditioned on $\mathcal{F}_i^c$, the event $\mathcal{F}_{i+1}$ cannot occur. Inductively, we then may write

$$\mathbb{P}\left[\mathsf{rank}_{\mathbb{F}_2}(\mathbf{B}) = k\right] = \prod_{i=0}^{k-1} \mathbb{P}\left[\mathcal{F}_{i+1}|\mathcal{F}_i\right] \tag{26}$$

with $\mathcal{F}_i = \emptyset$. We next lower bound each term in the product. To this end, recall that the fact that $\mathbf{B}_1, \ldots, \mathbf{B}_i$ are linearly independent implies that the $m \times i$ submatrix formed by these columns can be transformed into a matrix with the first $i$ columns forming an identity matrix, namely, the $i \times i$ identity matrix appears as a sub-block. Accordingly, this implies that *any* vector contained in the span of $\mathbf{B}_1, \ldots, \mathbf{B}_i$ can be represented as follows: its first $i$ entries can have arbitrary values, and the rest $m - i$ entries must be uniquely determined by the first $i$ entries. With this fact in mind, the $(i+1)$th column of $\mathbf{B}$ is linearly independent of the previous $i$ columns if and only if it is not spanned by these columns, or equivalently, if its last $m - i$ entries can be arbitrary. The probability of this being happen is clearly lower bounded by $1 - \alpha^{m-i}$ with $\alpha \triangleq \max(p, 1 - p)$. Combining the last observations, we obtain

$$\mathbb{P}\left[\mathsf{rank}_{\mathbb{F}_2}(\mathbf{B}) = k\right] \geq \prod_{i=0}^{k-1}(1 - \alpha^{m-i}) \tag{27}$$

$$= \prod_{\ell=m-k+1}^{m}(1 - \alpha^{\ell}). \tag{28}$$

The above result is general, but note that

$$\mathbb{P}\left[\mathsf{rank}_{\mathbb{F}_2}(\mathbf{B}) = k\right] \geq (1 - \alpha^{m-k+1})^k \tag{29}$$
$$\geq 1 - \min(1, k\alpha^{m-k+1}), \tag{30}$$

which concludes the proof.

## C   Proof of Proposition 1

In this section we analyze Algorithm 1, which extracts the clustering matrix when the clusters are disjoint. Pick $m$ elements uniformly at random from the set of elements $\mathcal{N}$. We denote this set by

$\mathcal{R}$. Perform all pairwise queries among these $m$ elements, resulting in a total of $\binom{m}{2}$ queries. We want to take $m$ large enough such that we have representatives of all clusters, namely, among these $m$ elements there will exists at least one element (representative) from each cluster. We next show that if $m \geq \frac{n}{n_{\min}} \log(kn^\varepsilon)$ than with probability decaying to zero polynomially in $n$, this is possible. Let $\mathcal{E}_\ell$ denote the event that no item in $\mathcal{R}$ appears in the $\ell$'th cluster. Then, we note that

$$\mathbb{P}\left[\bigcup_{\ell=1}^{k} \mathcal{E}_\ell\right] \leq k \cdot \max_{1 \leq \ell \leq k} \mathbb{P}\left[\mathcal{E}_\ell\right] \tag{31}$$

$$\leq k \cdot \left(1 - \frac{n_{\min}}{n}\right)^m \tag{32}$$

$$\leq k \cdot e^{-m\frac{n_{\min}}{n}} \tag{33}$$

$$\leq \frac{1}{n^\varepsilon}. \tag{34}$$

To wit, after the second stage of Algorithm 1 with high probability we found $k$ representatives $\mathcal{T}$ for the clusters. Finally, for the remaining $n - m$ items, we perform at most $k$ queries to decide which cluster they are in. Thus, the total number of quires is $k(n-m) + \left[\frac{n}{n_{\min}}\right]^2 \log^2(kn^\varepsilon)$.

## D   Proof of Theorem 1

In this section we analyze the performance of Algorithm 2, for the i.i.d. ensemble. The uniform ensemble is handled in the same way. In the first step of Algorithm 2, we pick $\mathcal{S}$ elements uniformly at random from $\mathcal{N}$ such that $m = |\mathcal{S}| > \mathsf{S}_{\text{i.i.d.}}$, where the latter is defined in (8). According to Lemma 6, this ensures that $\text{rank}(\mathbf{A}_\mathcal{S}) = k$ with high probability. We perform all pairwise queries among these $m$ elements, resulting in a total of $\binom{m}{2}$ queries. Then, in the second stage, we extract a valid membership of all chosen $m$ element by a simple rank factorization procedure. We denote by $\hat{\mathbf{A}}_\mathcal{S}$ the resultant rank factorized matrix, and we note that it might be not unique. Nonetheless, since $m > \mathsf{S}_{\text{i.i.d.}}$, we can find a subset of elements $\mathcal{T} \subseteq \mathcal{S}$ whose membership vectors form a basis of $\mathbb{R}^k$. Denote the $k \times k$ membership matrix corresponding to $\mathcal{T}$ by $\tilde{\mathbf{A}}_\mathcal{T}$. Then, in the third step of Algorithm 2, we query each of the remaining elements in $[n] \setminus \mathcal{S}$ with all the elements in $\mathcal{T}$. Accordingly, for any $i \in [n] \setminus \mathcal{S}$, let $\mathbf{c}_i$ be the $k$-length vector containing the $k$ queries of element $i$ with $\mathcal{T}$. Subsequently, given $\{\mathbf{c}_i\}_i$, we find the membership vector $\mathbf{m}_i$ of the $i$th element by solving $\tilde{\mathbf{A}}_\mathcal{T}\mathbf{m}_i = \mathbf{c}_i$, which form $k$ linearly independent equations in the $k$ variables. Thus, we can solve this system of equations uniquely to obtain the membership vector of $i$th element. Note that despite the fact that the second step of Algorithm 2 is not unique (and then $\hat{\mathbf{A}}_\mathcal{S}$ might be different from the true $\mathbf{A}_\mathcal{S}$), our algorithm will correctly recover the similarity matrix.

Indeed, let $\mathbf{B}_1$ and $\mathbf{B}_2$ be two solutions obtained by the rank factorization procedure, such that $\mathbf{B}_1\mathbf{B}_1^T = \mathbf{B}_2\mathbf{B}_2^T = \mathbf{A}_\mathcal{S}\mathbf{A}_\mathcal{S}^T$. Consider two elements, say, $\{1, 2\}$ whose membership vectors $\mathbf{m}_1$ and $\mathbf{m}_2$ are unknown after the second step of Algorithm 2. Since we query these elements with all the elements in $\mathcal{S}$, we must have the following set of equations

$$\begin{cases} \mathbf{B}_1\mathbf{m}_1 = \mathbf{c}_1, \\ \mathbf{B}_1\mathbf{m}_2 = \mathbf{c}_2, \end{cases} \qquad \begin{cases} \mathbf{B}_2\mathbf{m}_1 = \mathbf{c}_1, \\ \mathbf{B}_2\mathbf{m}_2 = \mathbf{c}_2. \end{cases} \tag{35}$$

Denote by $\hat{\mathbf{m}}_1$ and $\hat{\mathbf{m}}_2$ the solutions of $\mathbf{m}_1$ and $\mathbf{m}_2$, respectively, if $\mathbf{B}_1$ is the solution used. Similarly, let $\bar{\mathbf{m}}_1$ and $\bar{\mathbf{m}}_2$ be the solutions of $\mathbf{m}_1$ and $\mathbf{m}_2$, respectively, if $\mathbf{B}_2$ is the solution used. Then,

$$\hat{\mathbf{m}}_1^T\hat{\mathbf{m}}_2 = (\mathbf{B}_1^{-1}\mathbf{c}_1)^T\mathbf{B}_1^{-1}\mathbf{c}_2$$
$$= \mathbf{c}_1^T(\mathbf{B}_1^{-1})^T\mathbf{B}_1^{-1}\mathbf{c}_2$$
$$= \mathbf{c}_1^T(\mathbf{B}_1^T)^{-1}\mathbf{B}_1^{-1}\mathbf{c}_2$$
$$= \mathbf{c}_1^T(\mathbf{B}_1\mathbf{B}_1^T)^{-1}\mathbf{c}_2$$
$$= \mathbf{c}_1^T(\mathbf{B}_2\mathbf{B}_2^T)^{-1}\mathbf{c}_2$$
$$= \bar{\mathbf{m}}_1^T\bar{\mathbf{m}}_2, \tag{36}$$

---

**Algorithm 6** `Noisy Quantized Responses` The algorithm for extracting membership of elements via queries to oracle.

---

**Require:** Number of elements: $N$, number of clusters $k$, oracle responses $\mathcal{O}_{\text{quantized}}(i,j)$ for query $(i,j) \in \Omega$, where $i,j \in [N]$.

1: Choose a set $\mathcal{S}$ of elements drawn uniformly at random from $\mathcal{N}$, and perform all pairwise queries corresponding to these $|\mathcal{S}|$ elements.
2: Run Algorithm `NoisyInferSupport1` to infer $\langle \mathbf{A}_i, \mathbf{A}_j \rangle$ for each pair of entries $(i,j) \in \mathcal{S}$.
3: Extract the membership of all the $|\mathcal{S}|$ elements up to a permutation of the clusters.
4: Query each of the remaining $n - |\mathcal{S}|$ elements with all elements present in $\mathcal{S}$. Subsequently run algorithm `NoisyInferSupport2` for each query and solve for the membership vector of the unknown element.
5: Return the similarity matrix $\mathbf{A}\mathbf{A}^T$.

---

---

**Algorithm 7** `NoisyInferSupport1` The algorithm for inferring $\langle \mathbf{A}_i, \mathbf{A}_j \rangle$ for two fixed entries $(i,j) \in \mathcal{S}$.

---

**Require:** Set $\mathcal{S}$ where every pairwise value is observed, and indices $i,j \in \mathcal{S}$

1: Define $\Delta + 1$ numbers $E_\ell = (|\mathcal{S}| - 2)\left( (1-q)^2 - 2(1-2q)(1-q)\frac{\binom{k-\Delta}{\Delta}}{\binom{k}{\Delta}} + (1-2q)^2 \frac{\binom{k-2\Delta+\ell}{\Delta}}{\binom{k}{\Delta}} \right)$
   for $\ell = 0, 1, \ldots, \Delta$
2: Calculate $T_{ij} = \sum_{\substack{r \in \mathcal{S} \\ r \neq i,j}} \mathbb{1}[\mathbf{Y}_{ir} = 1 \cap \mathbf{Y}_{jr} = 1]$
3: Return $\arg\min_\ell |T_{ij} - E_\ell|$

---

which means that the inner products will be preserved. Hence we will get the same similarity matrix irrespective of the intermediate solution produced by the rank factorization which may be incorrect.

Finally, note that the number of queries needed in the above algorithm is $\binom{|\mathcal{S}|}{2} + k(n - |\mathcal{S}|)$, which concludes the proof.

# E   Proof of Theorem 2

In this section we analyze the Algorithm 6 for quantized noisy oracle, under the uniform ensemble.

Recall that we deal with the setting where the oracle responses are $\mathbf{Y}_{ij} = \mathcal{O}_{\text{quantized}}(i,j) = \mathcal{Q}(\mathbf{A}_i^T \mathbf{A}_j) \oplus W_{i,j}$, and we assume that $\mathbf{A}$ was generated according to the uniform ensemble with $k > 3\Delta$. Let $\mathcal{S}$ be a set drawn uniformly at random from $\mathcal{N}$, whose size will be determined in the sequel.

We next analyze the probability of error associated with Algorithm 6, by investigating each of its steps. Accordingly, in the first step of Algorithm 6, we observe $\mathbf{Y}_{ij}$ for all pairs $(i,j) \in \mathcal{S}$. Then, in the second step of Algorithm 6, using these $\binom{|\mathcal{S}|}{2}$ observations we infer $\langle \mathbf{A}_i, \mathbf{A}_j \rangle$, for any $(i,j) \in \mathcal{S}$. This is done using the procedure in Algorithm 7. To wit, at the end of the second step of Algorithm 6, we should have an exact estimate of $\mathbf{A}_{\mathcal{S}} \mathbf{A}_{\mathcal{S}}^T$ with high probability. In the following, we show that this is indeed correct.

---

**Algorithm 8** `NoisyInferSupport2` The algorithm for inferring $\langle \mathbf{A}_i, \mathbf{A}_j \rangle$ for $i \in \mathcal{S}, j \notin \mathcal{S}$.

---

**Require:** Set $\mathcal{S}$ where every pairwise value is observed, Indices $i \in \mathcal{S}, j \notin \mathcal{S}$.

1: Define $\Delta$ numbers $E_\ell = (|\mathcal{S}| - 1)\left( (1-q)^2 - 2(1-2q)(1-q)\frac{\binom{k-\Delta}{\Delta}}{\binom{k}{\Delta}} + (1-2q)^2 \frac{\binom{k-2\Delta+\ell}{\Delta}}{\binom{k}{\Delta}} \right)$
   for $\ell = 0, 1, \ldots, \Delta$
2: Calculate $T_{ij} = \sum_{\substack{r \in \mathcal{S} \\ r \neq i}} \mathbb{1}[\mathbf{Y}_{ir} = 1 \cap \mathbf{Y}_{jr} = 1]$
3: Return $\arg\min_\ell |T_{ij} - E_\ell|$

---

For a given pair $(i,j) \in \mathcal{S}$, we define a sequence of $(\Delta + 1)$ hypotheses $\{\mathcal{H}_\ell\}_{\ell=0}^{\Delta}$, where

$$\mathcal{H}_\ell : \quad \mathbf{A}_i^T \mathbf{A}_j = \ell \quad \text{for } \ell = 0, \dots, \Delta. \tag{37}$$

For a pair $(i,j) \in \mathcal{S}$, define

$$T_{i,j} \triangleq \sum_{\substack{r \in \mathcal{S} \\ r \neq i,j}} \mathbb{1}[\mathbf{Y}_{ir} = 1 \cap \mathbf{Y}_{jr} = 1]. \tag{38}$$

It is clear that each summand of $T_{i,j}$ is one if $\mathbf{Y}_{ir} = \mathbf{Y}_{jr} = 1$, and zero otherwise. We call the aforementioned event a *triangle* formed by the triplet $(i,j,r)$. Accordingly, the random variable $T_{i,j}$ simply counts/enumerate the number of triangles formed by a given pair $(i,j) \in \mathcal{S}$. As can be seen from Algorithm 7, the count $T_{i,j}$ is main quantity used to infer the value of $\mathbf{A}_i^T \mathbf{A}_j$. Accordingly, we need to understand its probabilistic behaviour. For simplicity of notation, in the following we denote by $\mathbb{P}_\ell(\cdot)$ and $\mathbb{E}_\ell(\cdot)$ the probability and the expectation operators conditioned on hypothesis $\mathcal{H}_\ell$ being true. Also, let $\mathbf{Q}_{ij} \triangleq \mathcal{Q}\left(\mathbf{A}_i^T \mathbf{A}_j\right)$. Then, for $k > 3\Delta$, it is an easy task to check that for a triplet $(i,j,r) \in [n]$, we have

$$\begin{aligned}
\mathbb{P}_\ell(\mathbf{Q}_{ir} = 1 \cap \mathbf{Q}_{jr} = 1) &= 1 - \mathbb{P}_\ell(\mathbf{Q}_{jr} = 0) \\
&\quad - \mathbb{P}_\ell(\mathbf{Q}_{ir} = 0) + \mathbb{P}_\ell(\mathbf{Q}_{ir} = 0 \cap {}_{jr} = 0) \\
&= 1 - 2\frac{\binom{k-\Delta}{\Delta}}{\binom{k}{\Delta}} + \frac{\binom{k-2\Delta+\ell}{\Delta}}{\binom{k}{\Delta}}.
\end{aligned} \tag{39}$$

In a similar fashion,

$$\begin{aligned}
\mathbb{P}_\ell(\mathbf{Q}_{ir} = 0 \cap \mathbf{Q}_{jr} = 1) &= \mathbb{P}_\ell(\mathbf{Q}_{ir} = 1 \cap \mathbf{Q}_{jr} = 0) \\
&= \mathbb{P}_\ell(\mathbf{Q}_{ir} = 0) - \mathbb{P}_\ell(\mathbf{Q}_{ir} = 0 \cap \mathbf{Q}_{jr} = 0) \\
&= \frac{\binom{k-\Delta}{\Delta}}{\binom{k}{\Delta}} - \frac{\binom{k-2\Delta+\ell}{\Delta}}{\binom{k}{\Delta}},
\end{aligned} \tag{40}$$

and,

$$\mathbb{P}_\ell(\mathbf{Q}_{ir} = 0 \cap \mathbf{Q}_{jr} = 0) = \frac{\binom{k-2\Delta+\ell}{\Delta}}{\binom{k}{\Delta}}. \tag{41}$$

Therefore, using the above results, we obtain by the law of total probability,

$$\begin{aligned}
\mathbb{P}_\ell(\mathbf{Y}_{ir} &= 1 \cap \mathbf{Y}_{jr} = 1) \\
&= (1-q)^2 \cdot \mathbb{P}_\ell(\mathbf{Q}_{ir} = 1 \cap \mathbf{Q}_{jr} = 1) \\
&\quad + q(1-q) \cdot \mathbb{P}_\ell(\mathbf{Z}_{ir} = 1 \cap \mathbf{Z}_{jr} = 1) \\
&\quad + q(1-q) \cdot \mathbb{P}_\ell(\mathbf{Q}_{ir} = 1 \cap \mathbf{Q}_{jr} = 1) \\
&\quad + q^2 \cdot \mathbb{P}_\ell(\mathbf{Q}_{ir} = 0 \cap \mathbf{Q}_{jr} = 0) \\
&= (1-q)^2 - 2(1-2q)(1-q)\frac{\binom{k-\Delta}{\Delta}}{\binom{k}{\Delta}} \\
&\quad + (1-2q)^2 \frac{\binom{k-2\Delta+\ell}{\Delta}}{\binom{k}{\Delta}}.
\end{aligned} \tag{42}$$

Accordingly, we obtain

$$\begin{aligned}
\mathbb{E}_\ell T_{i,j} &= \mathbb{E}_\ell \sum_{\substack{r \in \mathcal{S} \\ r \neq i,j}} \mathbb{1}[\mathbf{Y}_{ir} = 1 \cap \mathbf{Y}_{jr} = 1] \\
&= (|\mathcal{S}| - 2)\left( (1-q)^2 - 2(1-2q)(1-q)\frac{\binom{k-\Delta}{\Delta}}{\binom{k}{\Delta}} \right. \\
&\qquad \left. + (1-2q)^2 \frac{\binom{k-2\Delta+\ell}{\Delta}}{\binom{k}{\Delta}} \right).
\end{aligned} \tag{43}$$

Therefore for any two hypotheses $\mathcal{H}_\ell$ and $\mathcal{H}_{\ell'}$, we have

$$|\mathbb{E}_\ell T_{ij} - \mathbb{E}_{\ell'} T_{ij}| = \frac{(|\mathcal{S}| - 2)(1 - 2q)^2}{\binom{k}{\Delta}} \cdot$$

$$\cdot \left| \binom{k - 2\Delta + \ell}{\Delta} - \binom{k - 2\Delta + \ell'}{\Delta} \right|. \tag{44}$$

Now, given $(\mathbf{A}_i, \mathbf{A}_j)$ it is clear that the random variables $\mathbb{1}[\mathbf{Y}_{ir} = 1 \cap \mathbf{Y}_{jr} = 1]$, for $r \in \mathcal{S}, r \neq i, j$, are statistically independent and therefore we can apply standard concentration inequalities, such as, Chernoff's inequality, to show that the value of the random variable $T_{ij}$ is strongly concentrated around its mean. We state the following classical result (see, e.g., [10]).

**Lemma 7.** *[Chernoff's inequality] Let $(X_i)_{i=1}^n$ be a sequence of $n$ i.i.d. Bernoulli($p$) random variables. Then, for any $\mu > p$,*

$$\mathbb{P}\left[\frac{1}{n}\sum_{i=1}^n X_i > \mu\right] \leq e^{-n \cdot d_{\mathsf{KL}}(\mu||p)}. \tag{45}$$

Let $P_{\mathsf{error},1}^{(i,j)}$ designate the average probability of associated Algorithm 7, for a given pair $(i, j) \in \mathcal{S}$. Then, we have

$$P_{\mathsf{error},1}^{(i,j)} = \sum_{\ell=1}^{\Delta} \mathbb{P}\left(\mathcal{H}_\ell\right) \mathbb{P}_\ell \left[\mathrm{error}\right]$$

$$= \sum_{\ell=1}^{\Delta} \mathbb{P}\left(\mathcal{H}_\ell\right) \mathbb{P}_\ell \left[\min_{\ell' \neq \ell} |T_{ij} - \mathbb{E}_{\ell'} T_{ij}| < |T_{ij} - \mathbb{E}_\ell T_{ij}|\right]. \tag{46}$$

Now, note that

$$\mathbb{P}_\ell \left[\min_{\ell' \neq \ell} |T_{ij} - \mathbb{E}_{\ell'} T_{ij}| < |T_{ij} - \mathbb{E}_\ell T_{ij}|\right]$$

$$\leq \mathbb{P}_\ell \left[|T_{ij} - \mathbb{E}_\ell T_{ij}| > \frac{\min_{\ell' \neq \ell} |\mathbb{E}_\ell T_{ij} - \mathbb{E}_{\ell'} T_{ij}|}{2}\right] \tag{47}$$

where we have used the triangle inequality, i.e., $|a - b| \geq ||a| - |b||$, for any $a, b \in \mathbb{R}$. Then, using Lemma 7, we obtain

$$\mathbb{P}_\ell \left[|T_{ij} - \mathbb{E}_\ell T_{ij}| > \frac{\min_{\ell' \neq \ell} |\mathbb{E}_\ell T_{ij} - \mathbb{E}_{\ell'} T_{ij}|}{2}\right]$$

$$\leq 2 \cdot e^{-(|\mathcal{S}|-2)d_{\mathsf{KL}}(\alpha||\beta)} \tag{48}$$

where

$$\beta \triangleq (1 - q)^2 - 2(1 - 2q)(1 - q)\frac{\binom{k-\Delta}{\Delta}}{\binom{k}{\Delta}} + (1 - 2q)^2 \frac{\binom{k-2\Delta+\ell}{\Delta}}{\binom{k}{\Delta}},$$

and

$$\alpha \triangleq \beta$$

$$+ \frac{(1 - 2q)^2}{2\binom{k}{\Delta}} \min_{\ell' \neq \ell} \left| \binom{k - 2\Delta + \ell}{\Delta} - \binom{k - 2\Delta + \ell'}{\Delta} \right|.$$

To simplify the above result, recall Pinsker's inequality, which states that $d_{\mathsf{KL}}(p||q) \geq 2|p - q|^2$, for any $0 \leq p, q \leq 1$. Therefore,

$$d_{\mathsf{KL}}(\alpha||\beta) \geq 2|\alpha - \beta|^2$$

$$= \frac{(1 - 2q)^4}{2} \min_{\ell' \neq \ell} \frac{\left|\binom{k-2\Delta+\ell}{\Delta} - \binom{k-2\Delta+\ell'}{\Delta}\right|^2}{\binom{k}{\Delta}^2}$$

$$= \frac{(1 - 2q)^4}{2} \frac{\left|\binom{k-2\Delta+\ell}{\Delta} - \binom{k-2\Delta+\ell-1}{\Delta}\right|^2}{\binom{k}{\Delta}^2} \triangleq \eta(\ell). \tag{49}$$

Combining the above results with the fact that $\eta(\ell)$ is monotonically decreasing in $\ell$, we obtain

$$P_{\text{error},1}^{(i,j)} \leq 2 \cdot \max_{\ell \geq 1} e^{-(|\mathcal{S}|-2)\cdot\eta(\ell)} \tag{50}$$

$$= 2 \cdot e^{-(|\mathcal{S}|-2)\cdot\eta(1)}. \tag{51}$$

Therefore, at the end of the second stage of Algorithm 6, we will have an exact estimate of $\mathbf{A}_\mathcal{S}\mathbf{A}_\mathcal{S}^T$ if (51) is satisfied for all $(i,j) \in \mathcal{S}$. By the union bound, we obtain that the overall probability of error associated with second stage of Algorithm 6 is upper bounded by

$$P_{\text{error},1} \leq 2 \cdot \binom{|\mathcal{S}|}{2} \cdot e^{-(|\mathcal{S}|-2)\cdot\eta(1)}$$

$$\leq 2n^2 \cdot e^{-(|\mathcal{S}|-2)\cdot\eta(1)}. \tag{52}$$

Accordingly, taking $|\mathcal{S}| > \frac{1}{\eta(1)} \log(2n^{2+\varepsilon}) + 2$, for any $\varepsilon > 0$, is sufficient to bring the probability of error to at most $n^{-\varepsilon}$. Note that the above constraint on $|\mathcal{S}|$ expands to

$$|\mathcal{S}| > \frac{2\binom{k}{\Delta}^2 \log(2n^{2+\varepsilon})}{(1-2q)^4 \left[\binom{k-2\Delta+1}{\Delta} - \binom{k-2\Delta}{\Delta}\right]^2} + 2. \tag{53}$$

Next, given the exact estimate of $\mathbf{A}_\mathcal{S}\mathbf{A}_\mathcal{S}^T$ from the second step our algorithm, in the third step we extract the membership of all chosen $|\mathcal{S}|$ elements by a simple rank factorization procedure as in Algorithm 2. We denote the resultant rank factorized matrix by $\hat{\mathbf{A}}_\mathcal{S}$. Finally, we analyze the fourth step of Algorithm 6, in which for each index $j \notin \mathcal{S}$, we observe $\mathbf{Y}_{ij}$, for all $i \in \mathcal{S}$, and from these we would like to infer the leftover inner-products. This is done with the help of Algorithm 8 which we analyze in the sequel.

In fact the entire analysis of Algorithm 8 remains almost the same as that for Algorithm 7 (and therefore it is omitted), except now $T_{ij}$ is a sum of $|\mathcal{S}| - 1$ indicator random variables. Indeed, it can be shown that the average probability of error $P_{\text{error},2}^{(i,j)}$ associated with Algorithm 8 is upper-bounded as follows

$$P_{\text{error},2}^{(i,j)} \leq 2 \cdot e^{-(|\mathcal{S}|-1)\cdot\eta(1)}, \tag{54}$$

and accordingly, the overall probability of error associated with fourth stage of Algorithm 6 is upper bounded by

$$P_{\text{error},2} \leq 2 \cdot |\mathcal{S}| \cdot (n - |\mathcal{S}|) \cdot e^{-(|\mathcal{S}|-1)\cdot\eta(1)}$$

$$\leq 2n^2 \cdot e^{-(|\mathcal{S}|-1)\cdot\eta(1)}. \tag{55}$$

Accordingly, taking $|\mathcal{S}| > \frac{1}{\eta(1)} \log(2n^{2+\varepsilon}) + 1$, for any $\varepsilon > 0$, is sufficient to bring the probability of error to at most $n^{-\varepsilon}$. Thus, we may conclude that with high probability we have the exact values of $\mathbf{A}_i^T\mathbf{A}_j$, for all $i \notin \mathcal{S}$, and $j \in \mathcal{S}$. For each $i \notin \mathcal{S}$ we denote by $\mathbf{c}_i$ the $|\mathcal{S}|$ length vector containing the inner-products $\mathbf{A}_i^T\mathbf{A}_j$, for $j \in \mathcal{S}$.

Finally, the only thing that is left to do is solve for the membership vector of each of the $(n - |\mathcal{S}|)$ elements. This is done similarly as was done in Algorithm 2 (see the m 1). Specifically, from Lemma 5, we know that by taking $|\mathcal{S}| > \frac{\binom{k}{\Delta}}{f(\Delta,k)}[\log(ek) + c\log k] + c_2 \log n$, for some $c_1 > 1$ and $c_2 > 0$, the rows of $\mathbf{A}_\mathcal{S}$ form a basis of $\mathbb{F}_2^k$ with high probability. Note that (53) is a stringent condition, and thus Lemma 5 holds. Furthermore, were also able to observe that if the rows of $\mathbf{A}_\mathcal{S}$ formed a basis, then for an index $i \in \notin \mathcal{S}$, the set of values of $\{\mathbf{A}_i^T\mathbf{A}_j\}$, for all $j \in \mathcal{S}$ were enough to determine the vector $\mathbf{A}_i$. Indeed, given the resultant matrix $\hat{\mathbf{A}}_\mathcal{S}$ from the rank factorization step in the third step of Algorithm 6, the unknown membership vector $\mathbf{c}_j$ of the $j \notin \mathcal{S}$ element is found by solving $\hat{\mathbf{A}}_\mathcal{S}\mathbf{c}_j = \mathbf{s}_j$.

Finally, we conclude the proof by noting that the total number of observed entries is

$$|\Omega| = \binom{|\mathcal{S}|}{2} + |\mathcal{S}|(n - |\mathcal{S}|), \tag{56}$$

and that (53) is the stringent condition which ensures vanishing error probability.

**Algorithm 9** `Quantized Responses` The algorithm for extracting membership of elements via queries to oracle.

---

**Require:** Number of elements: $N$, number of clusters $k$, oracle responses $\mathcal{O}_{\mathsf{quantized}}(i,j)$ for query $(i,j) \in \Omega$, where $i,j \in [N]$.
 1: Choose a set $\mathcal{S}$ of elements drawn uniformly at random from $\mathcal{N}$, and perform all pairwise queries corresponding to these $|\mathcal{S}|$ elements.
 2: Run Algorithm `InferSupportsize` to infer $\|\mathbf{A}_i\|_0$, for $i \in \mathcal{S}$. Then, run Algorithm `InferIntersection1` to infer $\langle \mathbf{A}_i, \mathbf{A}_j \rangle$ for each pair of entries $i,j \in \mathcal{S}$.
 3: Extract the membership of all the $|\mathcal{S}|$ elements up-to a permutation of clusters.
 4: Run Algorithm `InferSupportsize2` to infer $\|\mathbf{A}_i\|_0$, for $i \notin \mathcal{S}$. Then, for $i \notin \mathcal{S}$, run Algorithm `InferIntersection2` to infer $\langle \mathbf{A}_i, \mathbf{A}_j \rangle$ for $j \notin \mathcal{S}$, and solve for the membership vector for all elements.
 5: Return the similarity matrix $\mathbf{A}\mathbf{A}^T$.

---

**Algorithm 10** `InferSupportsize1` The algorithm for inferring $\|\mathbf{A}_i\|_0$ for a fixed entry $i \in \mathcal{S}$.

---

**Require:** Set $\mathcal{S}$ where every pairwise value is observed, and index $i \in \mathcal{S}$.
 1: Define $\Delta$ numbers $E_\ell = (|\mathcal{S}| - 1)\left(1 - q - (1-2q)(1-p)^\ell\right)$ for $\ell = 0, 1, \ldots, k$
 2: Calculate $T_i = \sum_{\substack{r \in \mathcal{S} \\ r \neq i}} \mathbb{1}[\mathbf{Y}_{ir} = 1]$
 3: Return $\arg\min_\ell |T_i - E_\ell|$

---

# F  Proof of Theorem 3

In this section we analyze the Algorithm 9 for quantized noisy oracle, under the i.i.d. ensemble.

The main difference between Algorithm 6 and Algorithm 9 lies in the fact that for the i.i.d. ensemble we first need to infer the number of non-zero elements in every row of $\mathbf{A}$ (or, $\ell_0$ norm of every row), before proceeding with a similar analysis as in the proof of Theorem 2. As in Section E, we analyze the probability of error associated with Algorithm 9, by investigating each of its steps. Given a set $\mathcal{S}$, recall that the second step in this algorithm is to infer the number of non-zero elements $\mathbf{A}_i$, for $i \in \mathcal{S}$. This is done with the aid of Algorithm 10. For every index $i \in \mathcal{S}$, let

$$\mathcal{T}_i \triangleq \sum_{j \in \mathcal{S}: j \neq i} \mathbb{1}[\mathbf{Y}_{ij} = 1]. \tag{57}$$

Also, let $\{H_\ell\}_{\ell=0}^k$ be a sequence of Hypotheses defined as follows:

$$\mathcal{H}_\ell : \ \|\mathbf{A}_i\|_0 = \ell, \quad \ell = 0, \ldots, k. \tag{58}$$

As before, let $\mathbf{Q}_{ij} \triangleq \mathcal{Q}(\mathbf{A}_i^T \mathbf{A}_j)$. Then, it is clear that

$$\mathbb{P}_\ell [\mathbf{Q}_{ij} = 1] = 1 - (1-p)^\ell, \tag{59}$$

and therefore,

$$\begin{aligned}\mathbb{E}_\ell \mathcal{T}_i &= (|\mathcal{S}| - 1)\left[(1-q)(1-(1-p)^\ell) + q(1-p)^\ell\right] \\ &= (|\mathcal{S}| - 1)\left[1 - q - (1-2q)(1-p)^\ell\right].\end{aligned} \tag{60}$$

Accordingly, for any two different hypotheses $H_\ell$ and $H_{\ell'}$, we obtain

$$\begin{aligned}|\mathbb{E}_\ell \mathcal{T}_i - \mathbb{E}_{\ell'} \mathcal{T}_i| &= (|\mathcal{S}| - 1)(1 - 2q) \\ &\quad \cdot \left|(1-p)^\ell - (1-p)^{\ell'}\right|.\end{aligned} \tag{61}$$

---

**Algorithm 11** `InferSupportsize2` The algorithm for inferring $\|\mathbf{A}_i\|_0$ for a fixed entry $i \notin \mathcal{S}$.

---

**Require:** Set $\mathcal{S}$ where every pairwise value is observed, and index $i \notin \mathcal{S}$.
 1: Define $\Delta$ numbers $E_\ell = |\mathcal{S}|\left(1 - q - (1-2q)(1-p)^\ell\right)$ for $\ell = 0, 1, \ldots, k$
 2: Calculate $T_i = \sum_{r \in \mathcal{S}} \mathbb{1}[\mathbf{Y}_{ir} = 1]$
 3: Return $\arg\min_\ell |T_i - E_\ell|$

---

**Algorithm 12** `InferIntersection1` The algorithm for inferring $\langle \mathbf{A}_i, \mathbf{A}_j \rangle$ for two fixed entries $i, j \in \mathcal{S}$.

**Require:** Set $\mathcal{S}$ where every pairwise value is observed, and indices $i, j \in \mathcal{S}$.
1: **if** $\mathbf{Y}_{ij} = 0$ **then**
2:     Return 0
3: **else**
4:     Define $\Delta_{i,j}$ numbers $E_\ell = (|\mathcal{S}| - 2)\Big((1-q)^2 - (1-q)(1-2q)(1-p)^{\|\mathbf{A}_i\|_0} - (1-q)(1-$
       $2q)(1-p)^{\|\mathbf{A}_j\|_0} + (1-2q)^2(1-p)^{\|\mathbf{A}_i\|_0 + \|\mathbf{A}_j\|_0 - \ell}\Big)$ for $\ell = 1, \ldots, \Delta_{i,j}$
5:     Calculate $T_{i,j} = \sum_{\substack{r \in \mathcal{S} \\ r \neq i,j}} \mathbb{1}[\mathbf{Y}_{ir} = 1 \cap \mathbf{Y}_{jr} = 1]$
6:     Return $\arg\min_\ell |T_{i,j} - E_\ell|$

---

**Algorithm 13** `InferIntersection2` The algorithm for inferring $\langle \mathbf{A}_i, \mathbf{A}_j \rangle$ for $i \in \mathcal{S}, j \notin \mathcal{S}$.

**Require:** Set $\mathcal{S}$ where every pairwise value is observed, and indices $i \in \mathcal{S}, j \notin \mathcal{S}$.
1: **if** $\mathbf{Y}_{ij} = 0$ **then**
2:     Return 0
3: **else**
4:     Define $\Delta_{i,j}$ numbers $E_\ell = (|\mathcal{S}| - 1)\Big((1-q)^2 - (1-q)(1-2q)(1-p)^{\|\mathbf{A}_i\|_0} - (1-q)(1-$
       $2q)(1-p)^{\|\mathbf{A}_j\|_0} + (1-2q)^2(1-p)^{\|\mathbf{A}_i\|_0 + \|\mathbf{A}_j\|_0 - \ell}\Big)$ for $\ell = 1, \ldots, \Delta_{i,j}$
5:     Calculate $T_{i,j} = \sum_{\substack{r \in \mathcal{S} \\ r \neq i}} \mathbb{1}[\mathbf{Y}_{ir} = 1 \cap \mathbf{Y}_{jr} = 1]$
6:     Return $\arg\min_\ell |T_{i,j} - E_\ell|$

---

Now, given $\mathbf{A}_i$ it is clear that the random variables $\mathbb{1}[\mathbf{Y}_{ij} = 1]$, for $j \in \mathcal{S} \setminus \{i\}$, are statistically independent and therefore we can apply Lemma 7. Specifically, let $P_{\text{error},1}^{(i)}$ designate the average probability of associated Algorithm 10, for a given index $i \in \mathcal{S}$. Then, we have

$$P_{\text{error},1}^{(i)} = \sum_{\ell=0}^k \mathbb{P}(\mathcal{H}_\ell) \, \mathbb{P}_\ell \, [\text{error}]$$

$$= \sum_{\ell=1}^\Delta \mathbb{P}(\mathcal{H}_\ell) \, \mathbb{P}_\ell \left[ \min_{\ell' \neq \ell} |\mathcal{T}_i - \mathbb{E}_{\ell'} \mathcal{T}_i| < |\mathcal{T}_i - \mathbb{E}_\ell \mathcal{T}_i| \right]. \tag{62}$$

As in Appendix E (see eqs. (46)–(48)), we obtain

$$\mathbb{P}_\ell \left[ \min_{\ell' \neq \ell} |\mathcal{T}_i - \mathbb{E}_{\ell'} \mathcal{T}_i| < |\mathcal{T}_i - \mathbb{E}_\ell \mathcal{T}_i| \right]$$

$$\leq 2 \cdot e^{-(|\mathcal{S}|-1)\bar{\eta}(\ell)} \tag{63}$$

where

$$\bar{\eta}(\ell) \triangleq \frac{(1-2q)^2}{2} \left[ (1-p)^{\ell-1} - (1-p)^\ell \right]^2. \tag{64}$$

Combining the above results with the fact that $\bar{\eta}(\ell)$ is monotonically decreasing in $\ell$, we obtain

$$P_{\text{error},1}^{(i)} \leq 2 \cdot \max_{\ell \geq 1} e^{-(|\mathcal{S}|-1) \cdot \bar{\eta}(\ell)} \tag{65}$$

$$= 2 \cdot e^{-(|\mathcal{S}|-1) \cdot \eta(k)}. \tag{66}$$

Therefore, at the end of the first step in the second stage of Algorithm 9, we will have an exact estimate of $\|\mathbf{A}_i\|_0$, for $i \in \mathcal{S}$, if (66) is satisfied for all $i \in \mathcal{S}$. By the union bound, we obtain that the overall probability of error associated with this stage of Algorithm 9 is upper bounded by

$$P_{\text{error},1} \leq 2|\mathcal{S}| \cdot e^{-(|\mathcal{S}|-2) \cdot \eta(k)}$$

$$\leq 2n \cdot e^{-(|\mathcal{S}|-1) \cdot \bar{\eta}(k)}. \tag{67}$$

Accordingly, taking $|\mathcal{S}| > \frac{1}{\bar{\eta}(k)} \log(2n^{1+\varepsilon}) + 1$, for any $\varepsilon > 0$, is sufficient to bring the probability of error to at most $n^{-\varepsilon}$. Note that the above constraint on $|\mathcal{S}|$ expands to

$$|\mathcal{S}| > \frac{2 \log(2n^{1+\varepsilon})}{(1 - 2q)^2 \left[ (1 - p)^{k-1} - (1 - p)^k \right]^2} + 1. \tag{68}$$

After inferring the $\ell_0$-norm of each row, in the second step of Algorithm 9, we infer $\langle \mathbf{A}_i, \mathbf{A}_j \rangle$, for any $(i, j) \in \mathcal{S}$. This is done using the procedure in Algorithm 12. The analysis of this procedure is very similar to the analysis in Appendix E. In the following probabilities and expectations are evaluated conditioned on $\mathbf{A}_i^T \mathbf{A}_j = \ell$ and the values of $\|\mathbf{A}_i\|_0$ and $\|\mathbf{A}_j\|_0$. With some abuse of notation we denote these probabilities and expectations by $\mathbb{P}_\ell$ and $\mathbb{E}_\ell$, respectively. For a triplet $(i, j, r) \in [n]$, we have

$$\begin{aligned}
\mathbb{P}_\ell(\mathbf{Q}_{ir} = 1 \cap \mathbf{Q}_{jr} = 1) &= 1 - \mathbb{P}_\ell(\mathbf{Q}_{jr} = 0) \\
&\quad - \mathbb{P}_\ell(\mathbf{Q}_{ir} = 0) + \mathbb{P}_\ell(\mathbf{Q}_{ir} = 0 \cap \mathbf{Q}_{jr} = 0) \\
&= 1 - (1 - p)^{\|\mathbf{A}_i\|_0} - (1 - p)^{\|\mathbf{A}_j\|_0} \\
&\quad + (1 - p)^{\|\mathbf{A}_i\|_0 + \|\mathbf{A}_j\|_0 - \ell}.
\end{aligned} \tag{69}$$

In a similar fashion,

$$\begin{aligned}
\mathbb{P}_\ell(\mathbf{Q}_{ir} = 0 \cap \mathbf{Q}_{jr} = 1) &= \mathbb{P}_\ell(\mathbf{Q}_{ir} = 1 \cap \mathbf{Q}_{jr} = 0) \\
&= \mathbb{P}_\ell(\mathbf{Q}_{ir} = 0) - \mathbb{P}_\ell(\mathbf{Q}_{ir} = 0 \cap \mathbf{Q}_{jr} = 0) \\
&= (1 - p)^{\|\mathbf{A}_i\|_0} - (1 - p)^{\|\mathbf{A}_j\|_0} + (1 - p)^{\|\mathbf{A}_i\|_0 + \|\mathbf{A}_j\|_0 - \ell},
\end{aligned} \tag{70}$$

and finally,

$$\mathbb{P}_\ell(\mathbf{Q}_{ir} = 0 \cap \mathbf{Q}_{jr} = 0) = (1 - p)^{\|\mathbf{A}_i\|_0 + \|\mathbf{A}_j\|_0 - \ell}. \tag{71}$$

Therefore, using the above we obtain by the law of total probability,

$$\begin{aligned}
\mathbb{P}_\ell(\mathbf{Y}_{ir} = 1 \cap \mathbf{Y}_{jr} = 1) \\
&= (1 - q)^2 \cdot \mathbb{P}_\ell(\mathbf{Q}_{ir} = 1 \cap \mathbf{Q}_{jr} = 1) \\
&\quad + 2q(1 - q) \cdot \mathbb{P}_\ell(\mathbf{Q}_{ir} = 1 \cap \mathbf{Z}_{jr} = 1) \\
&\quad + q^2 \cdot \mathbb{P}_\ell(\mathbf{Q}_{ir} = 0 \cap \mathbf{Q}_{jr} = 0) \\
&= (1 - q)^2 - (1 - 2q)(1 - q)(1 - p)^{\|\mathbf{A}_i\|_0} \\
&\quad - (1 - 2q)(1 - q)(1 - p)^{\|\mathbf{A}_j\|_0} \\
&\quad + (1 - 2q)^2 (1 - p)^{\|\mathbf{A}_i\|_0 + \|\mathbf{A}_j\|_0 - \ell} \\
&\triangleq \mathcal{T}_{\text{th}}.
\end{aligned} \tag{72}$$

For a pair $(i, j) \in \mathcal{S}$, let us define

$$\Delta_{i,j} \triangleq \begin{cases} \min(\|\mathbf{A}_i\|_0, \|\mathbf{A}_j\|_0), & \text{if } \|\mathbf{A}_i\|_0 + \|\mathbf{A}_j\|_0 \leq k \\ \|\mathbf{A}_i\|_0 + \|\mathbf{A}_j\|_0 - k, & \text{if } \|\mathbf{A}_i\|_0 + \|\mathbf{A}_j\|_0 \geq k \end{cases}. \tag{73}$$

Accordingly, define a sequence of $\Delta_{ij}$ hypotheses $\{\bar{\mathcal{H}}_\ell\}_\ell$:

$$\bar{\mathcal{H}}_\ell : \quad \mathbf{A}_i^T \mathbf{A}_j = \ell \quad \text{for } \ell = 0, 1, \dots, \Delta_{i,j}. \tag{74}$$

Furthermore, define

$$\bar{T}_{i,j} \triangleq \sum_{\substack{r \in \mathcal{S} \\ r \neq i, j}} \mathbb{1}[\mathbf{Y}_{ir} = 1 \cap \mathbf{Y}_{jr} = 1]. \tag{75}$$

It follows that,

$$\mathbb{E}_\ell \bar{T}_{i,j} = (|\mathcal{S}| - 2) \mathcal{T}_{\text{th}}, \tag{76}$$

and thus, for any two hypotheses $\bar{\mathcal{H}}_\ell$ and $\bar{\mathcal{H}}_{\ell'}$, we have

$$|\mathbb{E}_\ell \bar{T}_{ij} - \mathbb{E}_{\ell'} \bar{T}_{ij}| = (|\mathcal{S}| - 2)(1 - 2q)^2$$
$$\cdot \left| (1-p)^{\|\mathbf{A}_i\|_0 + \|\mathbf{A}_j\|_0 - \ell} - (1-p)^{\|\mathbf{A}_i\|_0 + \|\mathbf{A}_j\|_0 - \ell'} \right|$$
$$\geq (|\mathcal{S}| - 2)(1 - 2q)^2 \left| (1-p)^{k-1} - (1-p)^k \right|. \tag{77}$$

Then, using the same machinery as in Appendix E (see eqs. (46)–(52)), it can be shown that the overall probability of error associated with second stage of Algorithm 9 is upper bounded by

$$\bar{P}_{\text{error},1} \leq 2 \cdot \binom{|\mathcal{S}|}{2} \cdot e^{-(|\mathcal{S}|-2) \cdot \tilde{\eta}(k)}$$
$$\leq 2n^2 \cdot e^{-(|\mathcal{S}|-2) \cdot \tilde{\eta}(k)} \tag{78}$$

where

$$\tilde{\eta}(k) \triangleq \frac{(1-2q)^4 \left[ (1-p)^{k-1} - (1-p)^k \right]}{2}.$$

Therefore, at the end of the second stage of Algorithm 9, if $|\mathcal{S}| > \frac{1}{\tilde{\eta}(k)} \log(2n^{2+\varepsilon}) + 2$, for any $\varepsilon > 0$, then we will have an exact estimate of $\mathbf{A}_\mathcal{S} \mathbf{A}_\mathcal{S}^T$ with probability of error to at most $n^{-\varepsilon}$. Note that the above constraint on $|\mathcal{S}|$ expands to

$$|\mathcal{S}| > \frac{2 \log(2n^{2+\varepsilon})}{(1-2q)^4 \left[ (1-p)^{k-1} - (1-p)^k \right]^2} + 2 \tag{79}$$

$$= \frac{2 \log(2n^{2+\varepsilon})}{p^2(1-2q)^4(1-p)^{2k-2}} + 2. \tag{80}$$

Next, given the exact estimate of $\mathbf{A}_\mathcal{S} \mathbf{A}_\mathcal{S}^T$ from the second step of our algorithm, in the third step we extract the membership of all chosen $|\mathcal{S}|$ elements by a simple rank factorization procedure as in Algorithm 2. We denote the resultant rank factorized matrix by $\hat{\mathbf{A}}_\mathcal{S}$. Finally, we analyze the fourth step of Algorithm 9, in which for each index $j \notin \mathcal{S}$, we observe $\mathbf{Y}_{ij}$, for all $i \in \mathcal{S}$, and from these we would like to infer the leftover inner-products. This is done with the help of Algorithms 11 and 13 which we analyze in the sequel.

In fact the entire analysis of Algorithms 11 and 13 remains almost the same. Indeed, in Algorithm 11 we infer $\|\mathbf{A}_i\|_0$, for $i \notin \mathcal{S}$. To this end, we define

$$\bar{\mathcal{T}}_i \triangleq \sum_{j \in \mathcal{S}} \mathbb{1}[\mathbf{Y}_{ij} = 1]. \tag{81}$$

It is evident that $\bar{\mathcal{T}}$ is very similar to (57), and thus, using the same steps as in (57)–(68), it can be shown that if

$$|\mathcal{S}| > \frac{2 \log(2n^{1+\varepsilon})}{(1-2q)^2 \left[ (1-p)^{k-1} - (1-p)^k \right]^2}. \tag{82}$$

then with overwhelming probability we correctly infer $\|\mathbf{A}_i\|_0$, for $i \notin \mathcal{S}$. Then, using the same arguments in (69)–(80), it can be shown that if

$$|\mathcal{S}| > \frac{2 \log(2n^{2+\varepsilon})}{p^2(1-2q)^4(1-p)^{2k-2}} + 1, \tag{83}$$

then Algorithm 13 succeeds, namely, with high probability, at the end of the fourth step of Algorithm 9, we have the exact values of $\mathbf{A}_i^T \mathbf{A}_j$, for all $i \notin \mathcal{S}$, and $j \in \mathcal{S}$. For each $i \notin \mathcal{S}$ we denote by $\mathbf{c}_i$ the $|\mathcal{S}|$ length vector containing the inner-products $\mathbf{A}_i^T \mathbf{A}_j$, for $j \in \mathcal{S}$. Note that (80) is the stringent condition among (68), (82), and (83), and thus if (80) holds the other conditions hold too.

Finally, the only thing that is left to do is solve for the membership vector of each of the $(n - |\mathcal{S}|)$ elements. This is done similarly as was done in Algorithm 2 (see the proof of Theorem 1). Specifically, from Lemma 6, we know that by taking $|\mathcal{S}| > k - 1 + \frac{1+\epsilon}{-\log \max(p, 1-p)} + c_2 \log n$, for some $\epsilon > 0$ and $c_2 > 0$, the rows of $\mathbf{A}_\mathcal{S}$ form a basis of $\mathbb{F}_2^k$ with high probability. Note that (80) is a stringent

**Algorithm 14** `Noisy Responses` The algorithm for extracting membership of elements via queries to oracle.

---

**Require:** Number of elements: $N$, number of clusters $k$, oracle responses $\mathcal{O}_{\mathsf{quantized}}(i,j)$ for query $(i,j) \in \Omega$, where $i,j \in [N]$.
 1: Choose a set $\mathcal{S}$ of elements drawn uniformly at random from $\mathcal{N}$, and perform all pairwise queries corresponding to these $|\mathcal{S}|$ elements. Compute $T_{ij} = \sum_{\substack{r \in \mathcal{S} \\ r \neq i,j}} \mathbb{1}[\mathbf{Y}_{ir} = 1 \cap \mathbf{Y}_{jr} = 1]$ for all $i,j \in \mathcal{S}$.
 2: Query the remaining $n - |\mathcal{S}|$ elements with all elements present in $\mathcal{S}$. Subsequently compute for all $i \in \mathcal{S}, j \notin \mathcal{S}$, $T_{ij} = \sum_{\substack{r \in \mathcal{S} \setminus \{x_j\} \\ r \neq i, x_j}} \mathbb{1}[\mathbf{Y}_{ir} = 1 \cap \mathbf{Y}_{jr} = 1]$ where $x_j$ is an arbitrarily selected element from $\mathcal{S}$ such that $x_j \neq i$
 3: Group all the $\binom{|\mathcal{S}|}{2} + |\mathcal{S}| \cdot (n - |\mathcal{S}|)$ counts $T_{ij}$ into $\Delta + 1$ groups such that the difference between any two intra-group points is smaller than the difference between any two inter-group points. If not possible, return NOT POSSIBLE.
 4: Order the groups by their value and label them by assigning the hypothesis $H_\ell$ to the $\ell^{th}$ group in the order.
 5: Assign $\langle \mathbf{A}_i, \mathbf{A}_j \rangle$ to be $\ell$ for queries $Q = (i,j)$ such that $T_{ij}$ belonged to the $\ell^{th}$ group.
 6: Extract the membership of all the $|\mathcal{S}|$ elements present in $\mathcal{S}$ by a rank factorization (may not be unique). Obtain $k$ linear independent vector from the solution space and represent them as a $\mathcal{T}$.
 7: Solve the membership vectors of all elements by solving the $k$ linearly independent equations obtained by getting the inner product with $\mathcal{T}$.
 8: Return the similarity matrix $\mathbf{A}\mathbf{A}^T$.

---

condition, and thus Lemma 6 holds. Furthermore, were also able to observe that if the rows of $\mathbf{A}_{\mathcal{S}}$ formed a basis, then for an index $i \in \notin \mathcal{S}$, the set of values of $\{\mathbf{A}_i^T \mathbf{A}_j\}$, for all $j \in \mathcal{S}$ were enough to determine the vector $\mathbf{A}_i$. Indeed, given the resultant matrix $\hat{\mathbf{A}}_{\mathcal{S}}$ from the rank factorization step in the third step of Algorithm 9, the unknown membership vector $\mathbf{c}_j$ of the $j \notin \mathcal{S}$ element is found by solving $\hat{\mathbf{A}}_{\mathcal{S}} \mathbf{c}_j = \mathbf{s}_j$.

Finally, we conclude the proof by noting that the total number of observed entries is

$$|\Omega| = \binom{|\mathcal{S}|}{2} + |\mathcal{S}| \cdot (n - |\mathcal{S}|), \tag{84}$$

while (80) ensures a vanishing error probability.

## G   Proof of Theorem 4

In this section, we analyze Algorithm 14. At the end of the second stage of Algorithm 14, we have access to all counts $T_{ij}$, for all pairs $(i,j) \in \mathcal{S}$, and $i \in \mathcal{S}, j \notin \mathcal{S}$. Suppose that these counts satisfy

$$\max_{\substack{T_{i_1 j_1} \in H_\ell \\ T_{i_2 j_2} \in H_\ell}} |T_{i_1 j_1} - T_{i_2 j_2}| \leq \delta \tag{85}$$

$$\min_{\substack{T_{i_1 j_1} \in H_\ell \\ T_{i_2 j_2} \in H_{\ell'} \\ \ell \neq \ell'}} |T_{i_1 j_1} - T_{i_2 j_2}| > 2\delta \tag{86}$$

where $T_{ij} \in H_\ell$ implies that $\mathbf{A}_i^T \mathbf{A}_j = \ell$. Now, according to the third stage of Algorithm 14, we group the counts $\{T_{ij}\}$ with the objective of forming $(\Delta + 1)$ clusters such that the count difference between any two intra-cluster points is less than the count difference between any two inter-cluster points. We next prove that counts belonging to two distinct hypotheses $H_\ell$ and $H_{\ell'}$ must also belong to different clusters. We prove this property by contradiction.

Indeed, the above claim can be wrong only if one of the following two situations happen: First, there are two clusters $\mathcal{A}$ and $\mathcal{B}$ both of which contain counts belonging to $H_\ell$ and $H_{\ell'}$. Denote the relevant counts in $\mathcal{A}$ by $a_\ell$ and $a_{\ell'}$, and the counts in $\mathcal{B}$ by $b_\ell$ and $b_{\ell'}$, where $a_\ell, b_\ell \in H_\ell$ and $a_{\ell'}, b_{\ell'} \in H_{\ell'}$. Then, according to (85)–(86), we must have $|a_\ell - a_{\ell'}| > |a_{\ell'} - b_{\ell'}|$, but this clearly contradicts the way the clusters were formed in the third step of Algorithm 14. The second situation

is when all counts belonging to $H_\ell$ and $H_{\ell'}$ are in the same cluster. However, since our objective is to find $(\Delta + 1)$ clusters, the counts in a particular hypotheses has to split into multiple clusters for this to happen. This implies, for example, that there exists three clusters $\mathcal{A}$, $\mathcal{B}$, and $\mathcal{C}$, and three hypotheses $H_\ell$, $H_{\ell'}$, and $H_{\hat{\ell}}$, such that that $\mathcal{A}$ and $\mathcal{B}$ contain counts belonging to $H_\ell$ only and $\mathcal{C}$ contains counts from $H_{\ell'}$ and $H_{\hat{\ell}}$. But then there exist counts in $\mathcal{C}$ whose difference is at least $2\delta$, whereas the maximum difference between counts in $\mathcal{A}$ and $\mathcal{B}$ is $\delta$ (since both contain counts from the same hypothesis), which again clearly contradicts the solution of the proposed algorithm. Therefore, we may conclude that, by construction, counts belonging to different hypotheses must belong to different clusters. Since we look for $(\Delta + 1)$ clusters, we exactly recover the clusters where each cluster corresponds to the counts of a particular hypothesis only. Moreover, we can correctly label the clusters as well because of the monotonicity of $\ell$ in the value of the counts belonging to hypothesis $H_\ell$ provided we have a valid solution by the algorithm.

In the following, we derive the sufficient conditions under which (85)-(86) are satisfied. First, note that in Algorithm 14 when computing the triangle counts for pairs $(i, j)$, such that $i \in \mathcal{S}$ and $j \notin \mathcal{S}$, we omit one arbitrarily picked element (denoted by $x_j$ where $x_j \neq i$) from $\mathcal{S}$. We do that because we want the expected value of the triangle count under the different hypotheses to be the same as in the case when $(i, j) \in \mathcal{S}$. Accordingly, recall (44). In order to satisfy (85)-(86), it is clear that $T_{ij}$ should deviate from its mean by at most $\min_{\ell, \ell': \ell \neq \ell'} \frac{|\mathbb{E}_\ell T_{ij} - \mathbb{E}_{\ell'} T_{ij}|}{6}$, which implies that

$$\delta = \frac{(|\mathcal{S}| - 2)(1 - 2q)^2}{3\binom{k}{\Delta}} \tag{87}$$
$$\cdot \left| \binom{k - 2\Delta + 1}{\Delta} - \binom{k - 2\Delta}{\Delta} \right|.$$

Then, using the same machinery as in Appendix A (see, eq. (46)–(52)), it can be shown that at the end of the third step of Algorithm 14, the overall probability of error is upper bounded by

$$P_{\text{error}} \leq 2n^2 \cdot e^{-(|\mathcal{S}| - 2) \cdot \bar{\eta}} \tag{88}$$

where

$$\bar{\eta} \triangleq \frac{(1 - 2q)^4}{18} \frac{\left| \binom{k - 2\Delta + 1}{\Delta} - \binom{k - 2\Delta}{\Delta} \right|^2}{\binom{k}{\Delta}^2}. \tag{89}$$

Accordingly, taking

$$|\mathcal{S}| > \frac{18\binom{k}{\Delta}^2 \log(2n^{2+\varepsilon})}{(1 - 2q)^4 \left[ \binom{k - 2\Delta + 1}{\Delta} - \binom{k - 2\Delta}{\Delta} \right]^2} + 2, \tag{90}$$

for any $\varepsilon > 0$, is sufficient to bring the probability of error to at most $n^{-\varepsilon}$.

It is evident that for the algorithm to return a valid solution, there must exist counts for all the $(\Delta + 1)$ hypotheses. We will show that this event happens with high probability under some conditions. For two indices $i, j \in [N]$, we have

$$\mathbb{P}(\mathbf{A}_i^T \mathbf{A}_j = \ell) = \frac{\binom{\Delta}{\ell}\binom{k - \Delta}{\Delta - \ell}}{\binom{k}{\Delta}}. \tag{91}$$

Then, it is clear that (91) is minimized when $\ell = \Delta$, in which case we have $\mathbb{P}(\mathbf{A}_i^T \mathbf{A}_j = \Delta) = \frac{1}{\binom{k}{\Delta}}$. If we only focus on an index $i \in \mathcal{S}$ (we are selecting an index in $\mathcal{S}$ because indices in $\mathcal{S}$ are queried with every other index in $[N]$), then let $U_{i,\ell}$ be the random variable which describes the number of indices (excluding $i$ itself) such that $\mathbf{A}_i^T \mathbf{A}_j = \ell$. It is clear that $U_{i,\ell}$ can be written as a sum of $(n - 1)$ i.i.d. binary random variables, and

$$\mathbb{E}(U_{i,\ell}) = \frac{(n - 1)\binom{\Delta}{\ell}\binom{k - \Delta}{\Delta - \ell}}{\binom{k}{\Delta}}. \tag{92}$$

Applying Chernoff's inequality once again, and taking a union bound over all $(\Delta + 1)$ hypotheses, we may conclude that if $n > 10\binom{k}{\Delta} \log n$, then $U_{i,\ell} > 0$, for all $\ell$, with high probability.

**Algorithm 15** `NoisyInferSupport1` The algorithm for inferring $\langle \mathbf{A}_i, \mathbf{A}_j \rangle$ for two fixed entries $i, j \in \mathcal{S}$.

---

**Require:** Set $\mathcal{S}$ where every pairwise value is observed, and indices $i, j \in \mathcal{S}$

1: Define $\Delta+1$ numbers $E_\ell = (|\mathcal{S}|-2)\left[1-2\mathbb{E}_\ell\left[Q\left(\frac{\mathbf{A}_j^T\mathbf{A}_r}{\sigma}\right)\right] - \mathbb{E}_\ell\left[Q\left(\frac{\mathbf{A}_j^T\mathbf{A}_r}{\sigma}\right)Q\left(\frac{\mathbf{A}_i^T\mathbf{A}_r}{\sigma}\right)\right]\right]$
   for $\ell = 0, 1, \ldots, \Delta$
2: Calculate $T_{ij} = \sum_{\substack{r \in \mathcal{S} \\ r \neq i,j}} \mathbb{1}[\mathbf{Y}_{ir} = 1 \cap \mathbf{Y}_{jr} = 1]$
3: Return $\arg\min_\ell |T_{ij} - E_\ell|$

---

**Algorithm 16** `NoisyInferSupport2` The algorithm for inferring $\langle \mathbf{A}_i, \mathbf{A}_j \rangle$ for $i \in \mathcal{S}, j \notin \mathcal{S}$.

---

**Require:** Set $\mathcal{S}$ where every pairwise value is observed, and indices $i \in \mathcal{S}, j \notin \mathcal{S}$.

1: Define $\Delta$ numbers $E_\ell = (|\mathcal{S}| - 1)\left[1 - 2\mathbb{E}_\ell\left[Q\left(\frac{\mathbf{A}_j^T\mathbf{A}_r}{\sigma}\right)\right] - \mathbb{E}_\ell\left[Q\left(\frac{\mathbf{A}_j^T\mathbf{A}_r}{\sigma}\right)Q\left(\frac{\mathbf{A}_i^T\mathbf{A}_r}{\sigma}\right)\right]\right]$
   for $\ell = 0, 1, \ldots, \Delta$
2: Calculate $T_{ij} = \sum_{\substack{r \in \mathcal{S} \\ r \neq i}} \mathbb{1}[\mathbf{Y}_{ir} = 1 \cap \mathbf{Y}_{jr} = 1]$
3: Return $\arg\min_\ell |T_{ij} - E_\ell|$

---

# H  Dithered Responses

In this section, we present our main result concerning dithered responses, i.e., $\mathcal{O}_{\text{dithered}}(i,j) = \mathcal{Q}\left(\mathbf{A}_i^T\mathbf{A}_j + Z_{i,j}\right)$, where $Z_{ij} \sim \text{Normal}(0, \sigma^2)$, independently over pairs $(i,j)$. Here, we consider the uniform ensemble only, but using the same techniques developed in this paper, the i.i.d. ensemble can be handled too. Let $Q(\cdot)$ denote the $Q$-function, namely, for any $x \in \mathbb{R}$, $Q(x) \triangleq \int_x^\infty \frac{1}{\sqrt{2\pi}}e^{-t^2/2}\mathrm{d}t$. Finally, for $\ell = 0, 1$, define

$$G_\ell(k, \Delta) \triangleq \mathbb{E}\left[Q\left(\frac{\mathbf{A}_1^T\mathbf{A}_3}{\sigma}\right)Q\left(\frac{\mathbf{A}_2^T\mathbf{A}_3}{\sigma}\right)\bigg| \mathbf{A}_1^T\mathbf{A}_2 = \ell\right] \tag{93}$$

where $\{\mathbf{A}_i\}_{i=1}^3$ are three statistically independent random vectors drawn from $T_k(\Delta)$. With these definitions, we are ready to state our main result.

**Theorem 6.** *Assume that* $\mathbf{A}$ *was generated according to the uniform ensemble. Then, there exists a polynomial-time algorithm, given in Algorithm 6, which with overwhelming probability, recovers the similarity matrix* $\mathbf{A}\mathbf{A}^T$*, using* $|\Omega| \geq \binom{|\mathcal{S}|}{2} + |\mathcal{S}| \cdot (n - |\mathcal{S}|)$ *queries, where for any* $\varepsilon > 0$*,*

$$|\mathcal{S}| > \frac{2\log(2n^{2+\varepsilon})}{|G_1(k,\Delta) - G_0(k,\Delta)|^2}. \tag{94}$$

*Proof of Theorem 6.*  The algorithm for this setting is the same as Algorithm 6, but with Algorithms 7 and 8 replaced with Algorithms 15 and 16. Accordingly, the main difference in the analysis compared to Appendix A is the computation of the statistics of the enumerators, and thus we omit some technical details. Specifically, recall that we assume that $\mathbf{A}$ was generated according to the uniform ensemble. Now, as before, we notice that for three distinct indices $(i, j, r) \in [n]$, we have

$$\mathbb{P}_\ell(\mathbf{Y}_{ir} = 1 \cap \mathbf{Y}_{jr} = 1) = 1 - 2 \cdot \mathbb{P}_\ell(\mathbf{Y}_{ir} = 0)$$
$$+ \mathbb{P}_\ell(\mathbf{Y}_{ir} = 0 \cap \mathbf{Y}_{jr} = 0). \tag{95}$$

Then, it is clear that

$$\mathbb{P}_\ell(\mathbf{Y}_{jr} = 0) = \mathbb{E}_\ell\left[Q\left(\frac{\mathbf{A}_j^T\mathbf{A}_r}{\sigma}\right)\right], \tag{96}$$

and

$$\mathbb{P}_\ell(\mathbf{Y}_{ir} = 0 \cap \mathbf{Y}_{jr} = 0)$$
$$= \mathbb{E}_\ell \left[ Q\left(\frac{\mathbf{A}_j^T \mathbf{A}_r}{\sigma}\right) Q\left(\frac{\mathbf{A}_i^T \mathbf{A}_r}{\sigma}\right) \right]. \tag{97}$$

It is also clear that (96) is independent of $\ell$ and $(i,j,r)$, while (97) depends on $\ell$ only. Therefore,

$$\mathbb{P}_\ell(\mathbf{Y}_{ir} = 1 \cap \mathbf{Y}_{jr} = 1) = 1$$
$$- \mathbb{E}_\ell \left[ 2Q\left(\frac{\mathbf{A}_j^T \mathbf{A}_r}{\sigma}\right) - Q\left(\frac{\mathbf{A}_j^T \mathbf{A}_r}{\sigma}\right) Q\left(\frac{\mathbf{A}_i^T \mathbf{A}_r}{\sigma}\right) \right]. \tag{98}$$

Next, as before, for a pair of indices $(i,j) \in \mathcal{S}$, define

$$T_{i,j} \triangleq \sum_{\substack{r \in \mathcal{S} \\ r \neq i,j}} \mathbb{1}[\mathbf{Y}_{ir} = 1 \cap \mathbf{Y}_{jr} = 1], \tag{99}$$

and thus,

$$\mathbb{E}_\ell T_{i,j} = (|\mathcal{S}| - 2)\left[ 1 - 2\mathbb{E}_\ell\left[ Q\left(\frac{\mathbf{A}_j^T \mathbf{A}_r}{\sigma}\right) \right] \right.$$
$$\left. - \mathbb{E}_\ell\left[ Q\left(\frac{\mathbf{A}_j^T \mathbf{A}_r}{\sigma}\right) Q\left(\frac{\mathbf{A}_i^T \mathbf{A}_r}{\sigma}\right) \right] \right]. \tag{100}$$

Accordingly for any two hypotheses $\mathcal{H}_\ell$ and $\mathcal{H}_{\ell'}$, we have

$$|\mathbb{E}_\ell T_{ij} - \mathbb{E}_{\ell'} T_{ij}|$$
$$= (|\mathcal{S}| - 2)\left| \mathbb{E}_\ell\left[ Q\left(\frac{\mathbf{A}_j^T \mathbf{A}_r}{\sigma}\right) Q\left(\frac{\mathbf{A}_i^T \mathbf{A}_r}{\sigma}\right) \right] \right.$$
$$\left. - \mathbb{E}_{\ell'}\left[ Q\left(\frac{\mathbf{A}_j^T \mathbf{A}_r}{\sigma}\right) Q\left(\frac{\mathbf{A}_i^T \mathbf{A}_r}{\sigma}\right) \right] \right| \tag{101}$$
$$\triangleq (|\mathcal{S}| - 2) \cdot \Gamma_{\ell,\ell'}. \tag{102}$$

Then, using the same machinery as in Appendix E (see eqs. (46)–(52)), it can be shown that the overall probability of error associated with second stage of Algorithm 9 for the dithered oracle is upper bounded by

$$P_{\text{error},1} \leq 2n^2 \cdot e^{-(|\mathcal{S}|-2)\cdot\frac{\Gamma_{1,0}^2}{2}}. \tag{103}$$

Therefore, at the end of the second stage of Algorithm 6, if $|\mathcal{S}| > \frac{2}{\Gamma_{1,0}^2} \log(2n^{2+\varepsilon}) + 2$, for any $\varepsilon > 0$, then we will have an exact estimate of $\mathbf{A}_\mathcal{S} \mathbf{A}_\mathcal{S}^T$ with probability of error to at most $n^{-\varepsilon}$. The other parts of the algorithm are handled in the same way (see eqs. (53)–(56), and thus omitted. We emphasize that as before, the over all query complexity $\binom{|\mathcal{S}|}{2} + |\mathcal{S}| \cdot (n - |\mathcal{S}|)$ is dominated by the above condition on $\mathcal{S}$. $\qquad\square$

## I Information-Theoretic Lower Bounds

In this section, we provide information-theoretic lower-bounds on the query complexity for exact recovery of the clustering matrix $\mathbf{A}$, associated with the scenarios considered in this paper. We denote by $\mathcal{H}_2(x)$ the binary entropy of $x \in (0,1)$, namely, $\mathcal{H}_2(x) \triangleq -x\log_2 x - (1-x)\log_2(1-x)$, and denote by $\star$ the binary convolution, i.e., $p \star q \triangleq (1-p)q + p(1-q)$. We have the following results proved in the sequel.

**Theorem 7.** *[i.i.d. Ensemble] Assume that* $\mathbf{A}$ *was generated accordingly to the i.i.d. ensemble with parameter* $p$. *Then, for any adaptive algorithm, in order to achieve* $\mathsf{P}_{\text{error}} \leq \delta$, *the necessary query complexity is*

- *For $\mathcal{O}_{\mathsf{direct}}$:*

$$|\Omega| \geq \frac{nk}{\log k} \cdot [\mathcal{H}_2(p) - \delta]. \tag{104}$$

- *For $\mathcal{O}_{\mathsf{quantized}}$:*

$$|\Omega| \geq nk \cdot \frac{\mathcal{H}_2(p) - \delta}{\mathcal{H}_2\left(q \star [1 - (1 - p^2)^k]\right) - \mathcal{H}_2(q)}. \tag{105}$$

- *For $\mathcal{O}_{\mathsf{dithered}}$:*

$$|\Omega| \geq \frac{nk \cdot [\mathcal{H}_2(p) - \delta]}{\mathcal{H}_2\left[\mathbb{E}Q\left(\frac{\mathbf{A}_1^T \mathbf{A}_2}{\sigma}\right)\right] - \mathbb{E}\mathcal{H}_2\left[Q\left(\frac{\mathbf{A}_1^T \mathbf{A}_2}{\sigma}\right)\right]}. \tag{106}$$

**Theorem 8.** *[Uniform Ensemble] Assume that $\mathbf{A}$ was generated accordingly to the uniform ensemble with parameter $\Delta$. Then, for any adaptive algorithm, in order to achieve $\mathsf{P}_{\mathsf{error}} \leq \delta$, the necessary query complexity is*

- *For $\mathcal{O}_{\mathsf{direct}}$:*

$$|\Omega| \geq nk \cdot \frac{\frac{1}{k} \log \binom{k}{\Delta} - \delta}{\log \Delta}. \tag{107}$$

- *For $\mathcal{O}_{\mathsf{quantized}}$:*

$$|\Omega| \geq nk \cdot \frac{\frac{1}{k} \log \binom{k}{\Delta} - \delta}{\mathcal{H}_2\left(q \star \frac{\binom{k-\Delta}{\Delta}}{\binom{k}{\Delta}}\right) - \mathcal{H}_2(q)}. \tag{108}$$

- *For $\mathcal{O}_{\mathsf{dithered}}$:*

$$|\Omega| \geq \frac{nk \cdot [\frac{1}{k} \log \binom{k}{\Delta} - \delta]}{\mathcal{H}_2\left[\mathbb{E}Q\left(\frac{\mathbf{A}_1^T \mathbf{A}_2}{\sigma}\right)\right] - \mathbb{E}\mathcal{H}_2\left[Q\left(\frac{\mathbf{A}_1^T \mathbf{A}_2}{\sigma}\right)\right]}. \tag{109}$$

### I.1 Proof of Theorem 7

#### I.1.1 Proof of Eq. 104

We consider the case where $\mathbf{A}$ was generated according to the i.i.d. ensemble. We observe $|\Omega|$ elements, drawn uniformly at random from the matrix $\mathbf{Y}$, where $\mathbf{Y}_{ij} = \mathcal{O}_{\mathsf{direct}}(i,j) = \mathbf{A}_i^T \mathbf{A}_j$. Let $\mathsf{P}_{\mathsf{error}}$ denotes the average probability of error associated with any estimator of $\mathbf{A}$ given the observations $\mathbf{Y}_\Omega$, namely, $\mathsf{P}_{\mathsf{error}} \triangleq \mathbb{P}\{\hat{\mathbf{A}}(\mathbf{Y}_\Omega) \neq \mathbf{A}\}$. We note that

$$H(\mathbf{A}) = H(\mathbf{A}|\Omega) \tag{110}$$

$$= I(\mathbf{A}; \mathbf{Y}_\Omega|\Omega) + H(\mathbf{A}|\mathbf{Y}_\Omega, \Omega) \tag{111}$$

$$\overset{\text{Fano}}{\leq} I(\mathbf{A}; \mathbf{Y}_\Omega|\Omega) + nk \cdot \lambda_{\mathsf{error}} \tag{112}$$

$$= H(\mathbf{Y}_\Omega|\Omega) - H(\mathbf{Y}_\Omega|\mathbf{A}, \Omega) + nk \cdot \mathsf{P}_{\mathsf{error}} \tag{113}$$

$$\overset{H(\mathbf{Y}_\Omega|\mathbf{A},\Omega)=0}{=} H(\mathbf{Y}_\Omega|\Omega) + nk \cdot \mathsf{P}_{\mathsf{error}} \tag{114}$$

where the inequality follows from Fano's inequality [10] which implies that

$$H(\mathbf{A}|\mathbf{Y}_\Omega) \leq \mathsf{P}_{\mathsf{error}} \cdot \log|\mathcal{A}| \leq nk \cdot \mathsf{P}_{\mathsf{error}} \tag{115}$$

where $\mathcal{A}$ is the set of all possible $n \times k$ binary matrices, and thus $|\mathcal{A}| = 2^{nk}$. Since $\mathbf{A}$ is an i.i.d. matrix with Bernoulli($p$) elements, we have $H(\mathbf{A}) = nk \cdot \mathcal{H}_2(p)$. Therefore, we obtain that

$$nk \cdot \mathcal{H}_2(p) \leq H(\mathbf{Y}_\Omega|\Omega) + nk \cdot \mathsf{P}_{\mathsf{error}}. \tag{116}$$

It is only left to upper bound the entropy $H(\mathbf{Y}_\Omega|\Omega) + nk \cdot \mathsf{P}_{\mathsf{error}}$. It is clear that

$$H(\mathbf{Y}_\Omega|\Omega) \leq |\Omega| \cdot \max_{i \neq j} H(\mathbf{A}_i^T \mathbf{A}_j) \tag{117}$$

$$\leq |\Omega| \cdot \log k \tag{118}$$

where the second inequality follows from the realization that $\mathbf{A}_i^T \mathbf{A}_j$ has a maximum value of $k$. Therefore, using (116), we obtain

$$nk \cdot \mathcal{H}_2(p) \leq |\Omega| \cdot \log k + nk \cdot \mathsf{P}_{\text{error}}. \tag{119}$$

Accordingly, to achieve $\mathsf{P}_{\text{error}} \leq \delta$, it is necessary that

$$|\Omega| \geq \frac{nk}{\log k}[\mathcal{H}_2(p) - \delta], \tag{120}$$

as claimed.

### I.1.2 Proof of Eq. 105

In this subsection we deal with the noisy quantized oracle, i.e., $\mathbf{Y}_{ij} = \mathcal{O}_{\text{quantized}}(\mathbf{A}_i^T \mathbf{A}_j) \oplus W_{ij}$. Similarly to (114), we have

$$H(\mathbf{A}) \leq H(\mathbf{Y}_\Omega|\Omega) - H(\mathbf{Y}_\Omega|\mathbf{A}, \Omega) + nk \cdot \mathsf{P}_{\text{error}} \tag{121}$$
$$= H(\mathbf{Y}_\Omega|\Omega) - |\Omega| \cdot \mathcal{H}_2(q) + nk \cdot \mathsf{P}_{\text{error}} \tag{122}$$

where we have used the fact that $H(\mathbf{Z}_\Omega|\mathbf{A}, \Omega) = |\Omega| \cdot \mathcal{H}_2(q)$. We next evaluate $H(\mathbf{Y}_\Omega|\Omega)$. Given $\Omega$, the $(i, j)$ element of $\mathbf{Y}$ is a Bernoulli random variable with success probability given by $q \star \beta_{ij}$, where $\beta_{ij} \triangleq \mathbb{P}\{\mathbf{A}_i^T \mathbf{A}_j > 0\}$, and $\star$ denotes the binary convolution. Now, note that for $i = j$,

$$\beta_{ii} = \mathbb{P}\{\|\mathbf{a}_i\|^2 > 0\} = 1 - \mathbb{P}\{\|\mathbf{a}_i\|^2 = 0\}$$
$$= 1 - (1 - p)^k, \tag{123}$$

and $i \neq j$,

$$\beta_{ij} = 1 - \mathbb{P}\{\mathbf{a}_i^T \mathbf{a}_j = 0\}$$
$$= 1 - (1 - p^2)^k. \tag{124}$$

Therefore,

$$H(\mathbf{Y}_\Omega|\Omega) \leq |\Omega| \cdot \max_{i,j} \mathcal{H}_2(q \star \beta_{ij})$$
$$\leq |\Omega| \cdot \mathcal{H}_2\left(q \star \left[1 - (1 - p^2)^k\right]\right). \tag{125}$$

Combining (122), (125), and the fact that $H(\mathbf{A}) = nk \cdot \mathcal{H}_2(p)$, we obtain

$$nk \cdot \mathcal{H}_2(p) \leq |\Omega| \cdot \mathcal{H}_2\left(q \star \left[1 - (1 - p^2)^k\right]\right)$$
$$- |\Omega| \cdot \mathcal{H}_2(q) + nk \cdot \mathsf{P}_{\text{error}}. \tag{126}$$

Accordingly, to achieve $\mathsf{P}_{\text{error}} \leq \delta$, it is necessary that

$$|\Omega| \geq nk \cdot \frac{\mathcal{H}_2(p) - \delta}{\mathcal{H}_2\left(q \star [1 - (1 - p^2)^k]\right) - \mathcal{H}_2(q)}, \tag{127}$$

as claimed.

### I.1.3 Proof of Eq. 106

We now consider the dithered oracle, where $\mathbf{Y}_{ij} = \mathcal{Q}(\mathbf{A}_i^T \mathbf{A}_j + Z_{ij})$, with $Z_{ij} \sim \text{Normal}(0, \sigma^2)$. Here, the analysis is very similar to the previous subsection. In particular, similarly to (122), we have

$$H(\mathbf{A}) \leq H(\mathbf{Y}_\Omega|\Omega) - H(\mathbf{Y}_\Omega|\mathbf{A}, \Omega) + nk \cdot \mathsf{P}_{\text{error}} \tag{128}$$
$$= H(\mathbf{Y}_\Omega|\Omega) - |\Omega| \cdot \mathbb{E}\mathcal{H}_2\left[Q\left(\frac{\mathbf{A}_1^T \mathbf{A}_2}{\sigma}\right)\right]$$
$$+ nk \cdot \mathsf{P}_{\text{error}}. \tag{129}$$

It is clear that given $\Omega$, the $(i, j)$ element of $\mathbf{Y}$ is a Bernoulli random variable with success probability $\mathbb{E}Q\left(\frac{\mathbf{A}_1^T \mathbf{A}_2}{\sigma}\right)$. Therefore, we obtain

$$H(\mathbf{Y}_\Omega|\Omega) \leq |\Omega| \cdot \mathcal{H}_2\left[\mathbb{E}Q\left(\frac{\mathbf{A}_1^T \mathbf{A}_2}{\sigma}\right)\right]. \tag{130}$$

Combining the above results and the fact that $H(\mathbf{A}) = nk\mathcal{H}_2(p)$, we may conclude that

$$nk\mathcal{H}_2(p) \leq |\Omega| \cdot \mathcal{H}_2 \left[ \mathbb{E}Q \left( \frac{\mathbf{A}_1^T \mathbf{A}_2}{\sigma} \right) \right]$$
$$- |\Omega| \cdot \mathbb{E}\mathcal{H}_2 \left[ Q \left( \frac{\mathbf{A}_1^T \mathbf{A}_2}{\sigma} \right) \right] + nk \cdot \mathsf{P}_{\mathsf{error}}. \tag{131}$$

Accordingly, to achieve $\mathsf{P}_{\mathsf{error}} \leq \delta$, it is necessary that

$$|\Omega| \geq \frac{nk \cdot [\mathcal{H}_2(p) - \delta]}{\mathcal{H}_2 \left[ \mathbb{E}Q \left( \frac{\mathbf{A}_1^T \mathbf{A}_2}{\sigma} \right) \right] - \mathbb{E}\mathcal{H}_2 \left[ Q \left( \frac{\mathbf{A}_1^T \mathbf{A}_2}{\sigma} \right) \right]}, \tag{132}$$

as claimed.

### I.2 Proof of Theorem 8

#### I.2.1 Proof of Eq. 107

We consider the case $\mathbf{A}$ where was generated according to the uniform ensemble, and the oracle response is $\mathbf{Y}_{ij} = \mathbf{A}_i^T \mathbf{A}_j$. Similarly as in (114), we have

$$H(\mathbf{A}) \leq H(\mathbf{Y}_\Omega | \Omega) + nk \cdot \mathsf{P}_{\mathsf{error}}. \tag{133}$$

For the uniform ensemble, note that $H(\mathbf{A}) = n \cdot \log \binom{k}{\Delta}$. Next, as in the previous subsection, note that

$$H(\mathbf{Y}_\Omega | \Omega) \leq |\Omega| \cdot \max_{i \neq j} H(\mathbf{A}_i^T \mathbf{A}_j) \tag{134}$$

$$\leq |\Omega| \cdot \log \Delta \tag{135}$$

where the second inequality follows from the realization that $\mathbf{A}_i^T \mathbf{A}_j$ has a maximum value of $\Delta$. Combining the above, we obtain

$$n \cdot \log \binom{k}{\Delta} \leq |\Omega| \cdot \log \Delta + nk \cdot \mathsf{P}_{\mathsf{error}}. \tag{136}$$

Accordingly, to achieve $\mathsf{P}_{\mathsf{error}} \leq \delta$, it is necessary that

$$|\Omega| \geq nk \cdot \frac{\frac{1}{k} \log \binom{k}{\Delta} - \delta}{\log \Delta}, \tag{137}$$

as claimed.

#### I.2.2 Proof of Eq. 108

We now deal with the noisy quantized oracle, i.e., $\mathbf{Y}_{ij} = \mathcal{O}_{\mathsf{quantized}}(\mathbf{A}_i^T \mathbf{A}_j) \oplus W_{ij}$. Similarly to (122), we have

$$H(\mathbf{A}) \leq H(\mathbf{Y}_\Omega | \Omega) - |\Omega| \cdot \mathcal{H}_2(q) + nk \cdot \mathsf{P}_{\mathsf{error}}. \tag{138}$$

It is clear that given $\Omega$, the $(i, j)$ element of $\mathbf{Y}$ is a Bernoulli random variable with success probability $\beta_{ij} \star q$, where $\beta_{ij} \triangleq \mathbb{P}\{\mathbf{a}_i^T \mathbf{a}_j > 0\}$. Note that for $i = j$,

$$\beta_{ii} = \mathbb{P}\{\|\mathbf{a}_i\|^2 > 0\} = 1, \tag{139}$$

while $i \neq j$,

$$\beta_{ij} = 1 - \mathbb{P}\{\mathbf{a}_i^T \mathbf{a}_j = 0\} = 1 - \frac{\binom{k-\Delta}{\Delta}}{\binom{k}{\Delta}}. \tag{140}$$

Therefore, using the above we obtain

$$H(\mathbf{Y}_\Omega | \Omega) \leq |\Omega| \cdot \mathcal{H}_2 \left( q \star \frac{\binom{k-\Delta}{\Delta}}{\binom{k}{\Delta}} \right). \tag{141}$$

Combining the above results and the fact that $H(\mathbf{A}) = n \cdot \log \binom{k}{\Delta}$, we may conclude that

$$n \cdot \log \binom{k}{\Delta} \leq |\Omega| \cdot \mathcal{H}_2 \left( q \star \frac{\binom{k-\Delta}{\Delta}}{\binom{k}{\Delta}} \right) - |\Omega| \cdot \mathcal{H}_2(q)$$
$$+ nk \cdot \mathsf{P}_{\text{error}}. \tag{142}$$

Accordingly, to achieve $\mathsf{P}_{\text{error}} \leq \delta$, it is necessary that

$$|\Omega| \geq nk \cdot \frac{\frac{1}{k} \log \binom{k}{\Delta} - \delta}{\mathcal{H}_2 \left( q \star \frac{\binom{k-\Delta}{\Delta}}{\binom{k}{\Delta}} \right) - \mathcal{H}_2(q)}, \tag{143}$$

as claimed.

### I.2.3 Proof of Eq. 109

We now consider the dithered oracle, where $\mathbf{Y}_{ij} = \mathcal{Q}(\mathbf{A}_i^T \mathbf{A}_j + Z_{ij})$, with $Z_{ij} \sim \mathsf{Normal}(0, \sigma^2)$. Here, the analysis is very similar to the Subsection I.1.3. In particular, similarly to (129), we have

$$H(\mathbf{A}) \leq H(\mathbf{Y}_\Omega | \Omega) - |\Omega| \cdot \mathbb{E}\mathcal{H}_2 \left[ Q \left( \frac{\mathbf{A}_1^T \mathbf{A}_2}{\sigma} \right) \right]$$
$$+ nk \cdot \mathsf{P}_{\text{error}}. \tag{144}$$

Also, similarly to (130), we have

$$H(\mathbf{Y}_\Omega | \Omega) \leq |\Omega| \cdot \mathcal{H}_2 \left[ \mathbb{E}Q \left( \frac{\mathbf{A}_1^T \mathbf{A}_2}{\sigma} \right) \right]. \tag{145}$$

Combining the above results and the fact that $H(\mathbf{A}) = n \log \binom{k}{\Delta}$, we conclude that

$$n \log \binom{k}{\Delta} \leq |\Omega| \cdot \mathcal{H}_2 \left[ \mathbb{E}Q \left( \frac{\mathbf{A}_1^T \mathbf{A}_2}{\sigma} \right) \right]$$
$$- |\Omega| \cdot \mathbb{E}\mathcal{H}_2 \left[ Q \left( \frac{\mathbf{A}_1^T \mathbf{A}_2}{\sigma} \right) \right] + nk \cdot \mathsf{P}_{\text{error}}. \tag{146}$$

Accordingly, to achieve $\mathsf{P}_{\text{error}} \leq \delta$, it is necessary that

$$|\Omega| \geq \frac{nk \cdot [\frac{1}{k} \log \binom{k}{\Delta} - \delta]}{\mathcal{H}_2 \left[ \mathbb{E}Q \left( \frac{\mathbf{A}_1^T \mathbf{A}_2}{\sigma} \right) \right] - \mathbb{E}\mathcal{H}_2 \left[ Q \left( \frac{\mathbf{A}_1^T \mathbf{A}_2}{\sigma} \right) \right]}, \tag{147}$$

as claimed.

## J   Worst Case Model: At Most 2 Clusters

In this section we prove the following special result for $\Delta = 2$.

**Theorem 9.** *Let $\mathcal{N}_i$ be the set of elements which belong to the $i$'th cluster, and assume that $\Delta = 2$. If, for every triplets of distinct clusters $p, q, r \in [k]$, we have $|\mathcal{N}_p \setminus \{\mathcal{N}_q \cup \mathcal{N}_r\}| > \alpha \cdot n$, for some $\alpha > 0$, then by using Algorithm 17, $\binom{T}{2} + T(n - T)$ queries are sufficient to recover the clusters, where $\alpha \cdot T = 3 \log k + \log n$.*

For ease of notation, we will say that an element tests *positive* with another element if the response to their query is 1 (i.e., they have one cluster in common). Otherwise, we will say they test *negative*. We will also say that a cluster is *maximal* if there does not exist any element that does not belong to the cluster but tests positive with every element in the cluster. The proof of Theorem 9 hangs on the following theorem.

**Theorem 10.** *Let $\mathcal{C}$ be a given clustering and let $\mathcal{N}_i$ be the set of elements which belong to the $i$'th cluster. If for every triplets of distinct clusters $p, q, r \in [k]$, we have $\mathcal{N}_p \setminus \{\mathcal{N}_q \cup \mathcal{N}_r\} \neq \phi$, then the ground truth clustering $\mathcal{C}$ is the only valid clustering that is consistent with the entire query matrix.*

**Algorithm 17** Worst-Case Quantized Responses for $\Delta = 2$ The algorithm for extracting membership of elements via queries to oracle for adversarial data.

---

**Require:** Number of elements: $N$, number of clusters $k$, oracle responses $\mathcal{O}_{\mathsf{quantized}}(i,j)$ for query $(i,j) \in \Omega$, where $i,j \in [N]$.
1: Choose a set $\mathcal{S}$ of elements drawn uniformly at random from $[N]$, and perform all pairwise queries corresponding to these $|\mathcal{S}|$ elements.
2: Construct a graph $\mathcal{G} = (\mathcal{V}, \mathcal{E})$ where the vertices are the $|\mathcal{S}|$ sampled elements. There exist an edge between elements $(i,j)$ only if they are determined to be similar by the oracle.
3: Construct the maximal cliques of the graph $\mathcal{G}$ such that all edges in $\mathcal{E}$ are covered and no three cliques intersect. Each maximal clique forms a cluster.
4: Query each of the remaining $n - |\mathcal{S}|$ elements with all elements present in $\mathcal{S}$. For each cluster, if an element is similar with all the elements in that particular cluster, then assign the element to that cluster.
5: Return all the clusters.

---

To prove this result we need the following lemma.

**Lemma 8.** *For a given clustering $\mathcal{C}$, if for every triplets of distinct clusters $p,q,r \in [k]$, we have $\mathcal{N}_p \setminus \{\mathcal{N}_q \cup \mathcal{N}_r\} \neq \phi$, then the clusters $\mathcal{N}_i$ are maximal.*

*Proof.* Proof of Lemma 8 We will prove this by contradiction. Suppose there exists a cluster $\mathcal{N}_i$ which is not maximal and there exist an element $x \notin \mathcal{N}_i$ such that $x$ tests positive with every element in $\mathcal{N}_i$. This is only possible if $\{x\} \cup \mathcal{N}_i \subset \mathcal{N}_j$ for some $j$ or if $\{x\} \cup \mathcal{N}_i \subset \mathcal{N}_j \cup \mathcal{N}_k$ (A partition of $\mathcal{N}_i$ into two sets $\mathcal{U}$ and $\mathcal{V}$ such that $\{x\} \cup \mathcal{U} \subset \mathcal{N}_j$ and $\{x\} \cup \mathcal{V} \subset \mathcal{N}_k$). Both these situations are not allowed according to our guarantees ($\mathcal{N}_i \setminus \{\mathcal{N}_j \cup \mathcal{N}_k\} \neq \phi$), which completes the proof. $\square$

We now prove Theorem 10.

*Proof of Theorem 10.* We will prove this result by induction on the number of clusters. Consider the base case of $k = 3$ where there are only three clusters say $\mathcal{N}_1, \mathcal{N}_2, \mathcal{N}_3$. Now the sets $\mathcal{N}_1 \setminus \{\mathcal{N}_2 \cup \mathcal{N}_3\}, \mathcal{N}_2 \setminus \{\mathcal{N}_1 \cup \mathcal{N}_3\}, \mathcal{N}_3 \setminus \{\mathcal{N}_1 \cup \mathcal{N}_2\}$ are non-empty and disjoint. In any different clustering $\tilde{\mathcal{C}}$, these three aforementioned sets have to belong to different clusters. Without loss of generality, assume that $\mathcal{N}_1 \setminus \{\mathcal{N}_2 \cup \mathcal{N}_3\} \subset \tilde{\mathcal{N}}_1$ and $\mathcal{N}_2 \setminus \{\mathcal{N}_1 \cup \mathcal{N}_3\} \subset \tilde{\mathcal{N}}_2$. In that case, it is easy to see that any element in $\mathcal{N}_1 \cap \mathcal{N}_2$ must belong to both $\tilde{\mathcal{N}}_1$ and $\tilde{\mathcal{N}}_2$ since it must test positive with elements in both $\mathcal{N}_1 \setminus \{\mathcal{N}_2 \cup \mathcal{N}_3\}$ and $\mathcal{N}_2 \setminus \{\mathcal{N}_1 \cup \mathcal{N}_3\}$. With this argument we get that the clustering $\tilde{\mathcal{C}}$ is the same as the clustering $\mathcal{C}$.

Now, assume that this lemma is true when there are $k$ clusters. Under this assumption, we will prove the statement of the lemma for $k+1$ clusters by contradiction. Assume that there exists a different clustering $\tilde{\mathcal{C}}$ such that there does not exist any $i,j \in [k]$ for which $\mathcal{N}_i = \tilde{\mathcal{N}}_j$. If $\mathcal{N}_1$ is a disjoint cluster that is $\mathcal{N}_1 \cap \mathcal{N}_j = \phi$ for all clusters $\mathcal{N}_j$, then all elements in $\mathcal{N}_1$ must belong to a disjoint cluster in $\tilde{\mathcal{C}}$ and we must have $\tilde{\mathcal{C}}$ to be the same as $\mathcal{C}$ by using the induction assumption. So now, we assume that no cluster $\mathcal{N}_i$ is disjoint. Assume that there exists some $i,j$ such that $\mathcal{N}_i \subset \tilde{\mathcal{N}}_j$. Since $\tilde{\mathcal{C}}$ is a valid clustering, hence all elements in $\tilde{\mathcal{N}}_j \setminus \mathcal{N}_i$ must test positive with all element in $\mathcal{N}_i$. This can happen only if 1) there exists some other cluster $\mathcal{N}_p$ such that $\mathcal{N}_i \cup \{\tilde{\mathcal{N}}_j \setminus \mathcal{N}_i\} \subset \mathcal{N}_p$ but this is not allowed since $\mathcal{N}_i \not\subset \mathcal{N}_p$. 2) If there exists two other clusters $\mathcal{N}_p$ and $\mathcal{N}_q$ such that $\mathcal{N}_i \cup \{\tilde{\mathcal{N}}_j \setminus \mathcal{N}_i\} \subset \mathcal{N}_p \cup \mathcal{N}_q$ but again this is not allowed since $\mathcal{N}_i \not\subset \mathcal{N}_p \cup \mathcal{N}_q$ (same argument as in proof of Lemma 8). So the previous assumption cannot happen and therefore there cannot exist some $i,j$ such that $\mathcal{N}_i \subset \tilde{\mathcal{N}}_j$ and by a similar argument there cannot exist $i,j$ such that $\tilde{\mathcal{N}}_i \subset \mathcal{N}_j$. Now, without loss of generality, assume that $\mathcal{N}_1 \cap \mathcal{N}_2 \neq \phi$. Hence there must exist some $\tilde{\mathcal{N}}_j$ such that $\tilde{\mathcal{N}}_j \cap \mathcal{N}_1 \cap \mathcal{N}_2 \neq \phi$. Let us denote one such element $x$ that belongs to $\tilde{\mathcal{N}}_j \cap \mathcal{N}_1 \cap \mathcal{N}_2$. Now there cannot exist an element $y \in \tilde{\mathcal{N}}_j \setminus \{\mathcal{N}_1 \cup \mathcal{N}_2\}$ because $y$ will test positive with $x$ but $x$ cannot belong to three clusters. Hence it must happen that $\tilde{C}_j \subset \mathcal{N}_1 \cup \mathcal{N}_2$. Now, consider two elements $z_1, z_2$ such that $z_1 \in \mathcal{N}_1 \setminus \mathcal{N}_2$ and $z_2 \in \mathcal{N}_2 \setminus \mathcal{N}_1$ such that $z_1$ and $z_2$ test negative. Such a pair of elements must exist otherwise the clusters $\mathcal{N}_1, \mathcal{N}_2$ will not be maximal according to Lemma 8. Now both the elements $z_1, z_2$ cannot belong to $\tilde{\mathcal{N}}_j$ since they test negative. On the other hand, both of them cannot

be outside $\tilde{\mathcal{N}}_j$ since if $x$ has to test positive with both $z_1, z_2$ then $x$ must belong to three clusters in $\tilde{\mathcal{C}}$ which is not allowed again. Hence, without loss of generality, assume that $z_1$ is contained in $\tilde{C}_j$. If $z_1$ only belongs to $\mathcal{N}_1$, then obviously no element from $\mathcal{N}_2 \setminus \mathcal{N}_1$ can belong to $\tilde{\mathcal{N}}_j$ (because $z_1$ will not test positive with that element) and therefore $\tilde{\mathcal{N}}_j \subset \mathcal{N}_1$ which is not allowed. Therefore, assume that $z_1$ also belongs to another cluster $\mathcal{N}_3$ and under this assumption, further assume that an element $z_3 \in \mathcal{N}_2 \cap \mathcal{N}_3$ is contained in $\tilde{\mathcal{N}}_j$ so that $\tilde{\mathcal{N}}_j \not\subset \mathcal{N}_1$. However, according to the guarantee that we are provided, there must exist an element $z_4 \in \mathcal{N}_1 \setminus \{\mathcal{N}_2 \cup \mathcal{N}_3\}$ and an element $z_5 \in \mathcal{N}_2 \setminus \{\mathcal{N}_1 \cup \mathcal{N}_3\}$. Now, neither of them can be included in $\tilde{\mathcal{N}}_j$ since $(z_4, z_3)$ and $(z_5, z_1)$ must test negative. If $(z_4, z_5)$ test negative, then this creates a contradiction since one of them have to be included in $\tilde{\mathcal{N}}_j$. Now if $(z_4, z_5)$ test positive, then one of $z_4$ and $z_5$ must belong to three clusters in $\tilde{\mathcal{C}}$ to satisfy the following constraints: $(z_4, x), (z_4, z_1), (z_5, z_3), (z_5, x), (z_4, z_5)$ test positive and $(z_4, z_3), (z_5, z_1)$ test negative $(z_1 \in \mathcal{N}_1 \cup \mathcal{N}_3$ and $z_5 \in \mathcal{N}_2 \setminus \{\mathcal{N}_1 \cup \mathcal{N}_3\}$ and similar for $(z_4, z_3))$ which is not allowed. Hence our initial assumption is incorrect and there cannot be a different clustering $\tilde{\mathcal{C}}$. $\square$

We are now ready to prove Theorem 9. The proof follows from the following three arguments.

1. Suppose we randomly sample a subset of elements $\mathcal{S}$ and let $\tilde{\mathcal{N}}_i = \mathcal{N}_i \cap \mathcal{S}$ be the set of elements in $\mathcal{S}$ which belong to the $i$'th cluster. A bad event is if there exist three distinct clusters $p, q, r \in [k]$ such that $\tilde{\mathcal{N}}_p \subset \tilde{\mathcal{N}}_q \cup \tilde{\mathcal{N}}_r$. For a particular triplet of clusters, the probability of this event to happen is clearly upper bounded by $(1-\alpha)^{|\mathcal{S}|} \leq e^{-\alpha|\mathcal{S}|}$. Taking a union bound over all triplets of clusters, the bad event will happen with probability at most $k^3 e^{-\alpha|\mathcal{S}|}$. Therefore, taking $\alpha \cdot |\mathcal{S}| = 3 \log k + \log n$ will make this probability at most $1/n$.

2. Now, from Theorem 10, it is easy to see that once we are given all the queries involving elements in $\mathcal{S}$, we are able to obtain the ground truth clustering and therefore all the clusters $\tilde{\mathcal{N}}_i$ produced by an algorithm that returns a valid clustering.

3. Finally, each element not in $\mathcal{S}$, will be queried with all elements in $\mathcal{S}$. If an element belongs to the $i$'th cluster, then obviously it will test positive with all elements in $\tilde{\mathcal{N}}_i$. If an element does not belong to the $i$'th cluster (say it belongs to the $j$'th cluster and $k$'th cluster) then it will not test positive with all elements in $\tilde{\mathcal{N}}_i$ (because of our guarantee). So we will recover the correct cluster every element belongs to.

It remains to show that Steps 2 and 3 in Algorithm 17 return a valid clustering if all the queries constrained to elements in $\mathcal{S}$ are provided. We know that all elements that belong to a particular cluster form a clique in the graph. We also know that all the edges can be covered by $k$ maximal cliques (the cliques can be overlapping) such that no three cliques intersect. Hence Step 3 of Algorithm 17 will return a valid clustering, which completes the the proof.

Finally, we notice that we can in fact show a necessary condition for the case of $\Delta = 2$, which almost coincide with Lemma 8, hinting that the above conditions might be also necessary.

**Lemma 9.** *Let $\mathcal{C}$ be a given clustering and let $\mathcal{N}_i$ be the set of elements which belong to the $i$th cluster. If for some pair of distinct clusters $p, q \in [k]$, $\mathcal{N}_p \subset \mathcal{N}_q$, then it is not possible to recover the ground truth clustering.*

*Proof of Lemma 9.* Consider a pair of clusters $\mathcal{N}_p, \mathcal{N}_q$ such that $\mathcal{N}_p \subset \mathcal{N}_q$. It is easy to see that it is impossible to determine which elements actually belong to the cluster $\mathcal{N}_q$ even if all possible query responses are provided. $\square$

# K   Proof of Theorem 5

We start this section by stating a conjecture which is the natural extension of Theorem 10 to any $\Delta > 0$.

**Conjecture 10.** *Let $\mathcal{C}$ be a given clustering and let $\mathcal{N}_i$ be the set of elements which belong to the $i$'th cluster. If for every ordered subset of $\Delta + 1$ distinct clusters $p_1, p_2, \ldots, p_{\Delta+1} \in [k]$, we have $\mathcal{N}_{p_1} \setminus \{\cup_{p_j \neq p_1} \mathcal{N}_{p_j}\} \neq \phi$, then the ground truth clustering $\mathcal{C}$ is the only valid clustering that is consistent with the entire query matrix.*

Unfortunately, we could not prove the above result, but rather the following weaker result.

**Theorem 11.** *Let $\mathcal{C}$ be a given clustering and let $\mathcal{N}_i$ be the set of elements which belong to the $i$'th cluster. If $\mathcal{N}_i \setminus \{\bigcup_{j \neq i} \mathcal{N}_j\}$ for all clusters $i \in [k]$, then the ground truth clustering $\mathcal{C}$ is the only valid clustering that is consistent with the entire query matrix.*

*Proof of Theorem 11.* Notice that the sets $\mathcal{N}_i \setminus \{\bigcup_{j \neq i} \mathcal{N}_j\}$, for all $i \in [k]$, are non-empty and disjoint. In any different clustering $\tilde{\mathcal{C}}$, the elements belonging to these aforementioned sets have to belong to different clusters. Without loss of generality, assume that $\mathcal{N}_i \setminus \{\bigcup_{j \neq i} \mathcal{N}_j \neq \mathcal{N}_i\} \subset \tilde{\mathcal{N}}_i$. In that case, for any subset $\mathcal{S} \subseteq [k]$, it is easy to see that any element in $\bigcap_{s \in \mathcal{S}} \mathcal{N}_s$ must belong to $\bigcap_{s \in S} \tilde{\mathcal{N}}_s$ since it must test positive with elements in $\mathcal{N}_i \setminus \{\bigcup_{j \neq i} \mathcal{N}_j \neq \mathcal{N}_i\}$ for all $i \in \mathcal{S}$ and tests negative with elements in $\mathcal{N}_i \setminus \{\bigcup_{j \neq i} \mathcal{N}_j \neq \mathcal{N}_i\}$ for all $i \notin \mathcal{S}$. With this argument we get that the clustering $\tilde{\mathcal{C}}$ is the same as the clustering $\mathcal{C}$. $\square$

We are now in a position to prove Theorem 5. The proof hangs on the following three arguments.

1. Suppose we randomly sample a subset of elements $\mathcal{S}$ and let $\tilde{\mathcal{N}}_i = \mathcal{N}_i \cap \mathcal{S}$ be the set of elements in $\mathcal{S}$ which belong to the $ith$ cluster. A bad event is if there exists a cluster $i \in [k]$ such that $\tilde{\mathcal{N}}_i \setminus \{\bigcup_{j:j \neq i} \tilde{\mathcal{N}}_j\} = \phi$. For a particular cluster, the probability of this event is upper bounded by $(1-\alpha)^{|\mathcal{S}|} \leq e^{-\alpha|\mathcal{S}|}$. Taking a union bound over all clusters, the bad event will happen with probability at most $ke^{-\alpha|\mathcal{S}|}$. Therefore, taking $\alpha \cdot |\mathcal{S}| = \log k + \log n$, will make this probability at most $1/n$.

2. Now, from Theorem 11, it is easy to see that once we are given all the queries involving elements in $\mathcal{S}$, we are able to obtain the ground truth clustering and therefore all the clusters $\tilde{\mathcal{N}}_i$ by an algorithm that returns a valid clustering. If the clusters are maximal, then Step 3 in Algorithm 3 (a slightly modified version of Algorithm 17) returns a valid and unique clustering.

3. Finally, each element not in $\mathcal{S}$ will be queried with all elements in $\mathcal{S}$. If an element belongs to the $i$'th cluster, then obviously it will test positive with all elements in $\tilde{\mathcal{N}}_i$. If an element does not belong to the $i$'th cluster then it will not test positive with all elements in $\tilde{\mathcal{N}}_i$ (because of our guarantee). So we will recover the correct cluster every element belongs to.

## L  Experiments on Simulated Data

We conduct in-depth simulations of the proposed techniques over synthetic data. We focus on the uniform ensemble and the quantized noisy oracle $\mathcal{O}_{\mathsf{quantized}}$. Recall that in our proposed algorithms (see, e.g., Algorithm 6), we make $|\Omega| = \binom{|\mathcal{S}|}{2} + |\mathcal{S}|(n - |\mathcal{S}|)$ queries, and for every query $(i,j) \in \Omega$, we infer using the count $T_{ij}$ the unquantized value $\mathbf{A}_i^T \mathbf{A}_j$. Accordingly, for evaluation, we investigate the amount of incorrect inferences made by our algorithm. It is only possible to recover the original matrix $\mathbf{A}$ only if all the inferences are correct (using Algorithm 4). Fig. 2a presents the log-query complexity ($\log_e |\Omega|$) as a function of the number of items $n$, for $\Delta = 2$, $k = 8$, and $q = 0$. We compare the simulated performance of Algorithm 6 with the theoretical lower and and upper bounds in Theorems 2 and 8, respectively. It can be seen that our theoretical upper bound follows closely the numerically evaluated performance of Algorithm 6.Fig. 2b shows $\log |\Omega|$ as a function of the number of clusters $k$, for $\Delta = 2$, $n = 4000$, and $q = 0$, and the same conclusions as above remain true. Then, in Figs. 2c–2e we consider the noisy scenario with $q$ controlling the "amount" of noise. We first assume that the value of $q$ is known. Specifically, in Fig. 2c we present $\log |\Omega|$ as a function of the noise parameter $q$, for $n = 2000$, $k = 7$, and $\Delta = 2$. Again, it can be seen that our theoretical upper bound match the simulated performance of Algorithm 6. We notice that the effect of the noise on the query complexity is not drastic, which imply that the proposed algorithm is robust. To illustrate the underlying mechanism of Algorithm 6, in Fig. 2d we present the amount of correct and wrong inferences occurred at the end of the second step of Algorithm 6, for $n = 1000$, $k = 8$, $\Delta = 2$. In this figure, we took $|\mathcal{S}| = 400$, which is the sufficient size for recovery in the noiseless case (but not for the noisy regime). It can be seen that the number of wrong inferences grows moderately up to $q \approx 0.1$, and then the effect of choosing an insufficient $|\mathcal{S}|$ becomes more

(a) Query complexity $\log|\Omega|$ of Algorithm 6 (blue) as a function of $n$, for $\Delta = 2$, $k = 8$, and $q = 0$. The green and red curves represent the lower and upper bound in Thms. 2 and 8.

(b) Query complexity $\log|\Omega|$ of Algorithm 6 (blue) as a function of $k$, for $\Delta = 2$, $n = 4000$, and $q = 0$. The green and red curves represent the lower and upper bound in Thms. 2 and 8.

(c) Query complexity $\log|\Omega|$ of Algorithm 6 (blue) as a function of $q$, for $\Delta = 2$, $k = 7$, and $n = 2000$. The green and red curves represent the lower and upper bound in Thms. 2 and 8.

(d) Number of correct/wrong inferences as a function of $q$, for $n = 1000$, $k = 8$, $\Delta = 2$, and $|\mathcal{S}| = 400$.

(e) Histogram of the counts statistic $\{T_{ij}\}$ used for inferring in Algorithm 14, for $n = 2000$, $k = 2$, and $\Delta = 2$.

Figure 2: Results of our techniques on simulated datasets.

severe. This suggests the potential application of our algorithms also when partial, rather than exact, recovery is the performance criterion. Finally, we illustrate how Algorithm 14 works in the absence of noise. Specifically, in Fig. 2e, we provide a histogram of the counts $T_{ij}$ defined in Algorithm 14, for $\Delta = 2$, $k = 7$, $n = 2000$ and $q = 0$. It is evident that the data can be separated into three groups (recall the third step of Algorithm 14) and therefore it is possible to infer correctly $\mathbf{A}_i^T \mathbf{A}_j$ for all pairwise queries.