[Reviews · NeurIPS 2019]

Reviewer 1



This is a well-written theory paper about a novel problem that has not been addressed in this form in earlier literature. The main novelty is the combination of pairwise queries and overlapping clusters. The results are solid. A summary (table, etc) of the main complexity results for different oracle types would make a nice addition. About related work: The authors might also be interested in an earlier HCOMP 2015 paper by Zou et al where a very similar problem using relative comparison queries is considered. Also, Bonchi et al (2013) discuss ideas to combine correlation clustering with overlapping clusters that may also be of interest to the authors. (Note that I am *not* asking to cite these papers, but merely pointing them out as potentially interesting related work.) References: Francesco Bonchi, Aristides Gionis, Antti Ukkonen: Overlapping correlation clustering. Knowl. Inf. Syst. 35(1): 1-32 (2013) James Y. Zou, Kamalika Chaudhuri, Adam Tauman Kalai: Crowdsourcing Feature Discovery via Adaptively Chosen Comparisons. HCOMP 2015: 198- _______________ Update after rebuttal: Thank you for the response! I have nothing to add at this point.

Reviewer 2



The authors propose the problem of clustering with overlapping under the semi-supervised framework, or more specifically, using same cluster queries. Flat clustering with same cluster queries has received much attention recently. The authors take a step forward to the case of clustering with overlapping, which is more general comparing to the original problem. I think the importance of this new problem is fairly stated in the paper. The related works section is well-written which gives a clear idea how this work is different from prior works. The main idea of all the algorithms in this paper is to find a (possibly small) set of representatives first using all possible pair-wise queries from a larger set drawn uniformly at random. Then recover the memberships for the rest elements depending on the result of the first stage. Although the similar idea has appeared in prior works (i.e. [MS17]), which is also cited by the authors, applying it for the case of overlapping clusters with modification for numbers of settings is still novel. The theoretical analysis for this paper seems to be correct, although I did not read all the details in the supplement. The first contribution of this paper is providing the condition of uniqueness of optimal solution in various settings. I think this is an important step for the follow up works tackling this problem. Various settings for this problem are discussed in this paper. Both upper bounds along with algorithms and lower bounds on query complexity are provided, which is another significant contribution. However, the authors do not make comparison for their upper and lower bounds. It would be more clearly for readers if some remarks of this comparison can be made. Moreover, the computational complexity of these algorithms is not directly stated. The computational complexity is also a critical attribute when we want to judge algorithms. Finally, although the authors give results for both worst-case and model-based, they make neither comparison nor discussion on the query complexity. Intuitively the query complexity for the worst-case scenario should be much higher than the model-based case. However, the results do not seem to match this intuition if \alpha is some constant. The authors should have some discussion on the scale of \alpha and compare the query complexity of worst-case and model-based case. For the experiment, the authors show that on real-world data, their algorithms require much less queries then their upper bound. Nevertheless, no other method is compared. For example, the authors mention in the introduction that their algorithm should work better then naively apply the low-rank matrix completion, but no experiment support this statement. Also, from the synthetic data in the supplement, there seems to be a huge gap for the upper and lower bound. Since the authors take a log-scale in y-axis it is hard for me to tell whether the upper bound or lower bound is tight or not. It would be great if some discussion can be made for this point. At last, in the reproducibility response the authors claim that they provide the source codes, but I can not find it in the supplement. [MS17] A. Mazumdar and B. Saha. Clustering with noisy queries. In Advances in Neural Information Processing Systems (NIPS) 31, 2017. ==================================================== Update after rebuttal: I thank the authors for their response. It clarify all my concerns pretty well. Hence I slightly raise my score to 7 for this submission. Hope that all these nice explanation in the response can be seen in the final version of this paper.

Reviewer 3



The authors extend the study of clustering with queries to the scenarios where the clusters may overlap. There has been previous work regarding the query complexity of clustering using same-cluster queries. However, this is the first work that deals with situations that a point can belong to multiple clusters. Multiple types of oracles has been considered including + Direct response oracle where given two points, the oracle outputs the number of clusters that the two points belong to simultaneously (also called the similarity of the two points). + Quantized oracle where the output is just whether there exist a cluster where the two points belong to it simultaneously. Noisy versions of this oracle is also considered where the oracle's output is flipped with some fixed probability. + Dithered oracle where the oracle first adds a Gaussian noise to the similarity of the two given points and then outputs the quantized version. The authors propose methods that recover the true clustering (up to a permutation) with high probabilities. In order to avoid the identifiability issues, the authors consider assumptions under which knowing the similarity matrix is enough to reconstruct the clustering assignments (at least with high probability). I think the paper is well-written and the setting is very interesting. Adding short discussion regarding the tightness of the upper bounds for each scenario (and the possible room for improvement) would help as there are various variables involved in the bounds. Some questions. + In Theorem 5, the given query complexity does not depend on Delta but intuitively it should be higher for larger Delta. Can you clarify? + In Theorem 5, T is logarithmic in k and n...so in the special case where the clusters are non-overlapping we seem to be able to recover the clustering using less than nk queries...which should be impossible. What am I missing? + Theorem 2 has exponential dependence on Delta. Is that really necessary? + I did not fully follow the role of |S| in the experiments; is there an intuition why the theoretical value does not work as well? + In the last experiment (unquantized query responses) a particular heuristic has been used (e.g., selecting 5 movies, etc.). Why do we do that (and not follow the proposed algorithm)? Will the result be different if we did so? how much? == After reading other reviews and the authors' response I see no reason to change my score.

[Author Response · NeurIPS 2019]

First of all we would like to thank all the reviewers for their careful reading and constructive comments. Please see the responses below.

**Reviewer #1**: Adding a table summarizing the sample complexity results for the different oracle types is a great suggestion. We also thank the reviewer for pointing out the two relevant references, which we will add in the revised version.

**Reviewer #2**: **1)** Comp. Complexity: In general, the computational complexities of our algorithms are roughly of the order of $O(|\Omega| + |\mathcal{S}|^3)$, dominated by querying $|\Omega|$ random samples and applying a rank factorization on the gram matrix $\mathbf{A}_{\mathcal{S}}\mathbf{A}_{\mathcal{S}}^T$. We will add the computational complexities of each algorithm in the revised version.

**2)** Tightness of Bounds: Again, generally speaking, we can show that the scaling of the upper and lower bounds with respect to the various parameters (such as $\Delta$, $k$, and $n$) is similar up to polylog factors. We will discuss the tightness of our IT lower and algorithmic upper bounds in the final version.

**3)** Matrix completion: Indeed, as was mentioned in the introduction, the sample complexity in our approach is smaller than those obtained using matrix completion when we deal with *unquantized* responses. For quantized responses, however, as far as we know, current matrix completion literature handle *dithered* quantized responses only (namely, adding continuous noise, e.g., Gaussian, before quantization). In our setting, we deal with (possibly noisy) quantized responses without any continuous dithering, and so matrix completion results cannot be used. In fact, without dithering matrix completion algorithms will fail on quantized data (see, Davenport, M. et. al., "1-Bit Matrix Completion"), as they do not exploit the discrete structure of the data, which is the main source for the success of our algorithms. Nonetheless, following the reviewer's comment, for direct responses, we can add experiments comparing our results with matrix completion algorithms if desired.

**4)** Scale of $\alpha$ and worst vs. average: Depending on the dataset, the scaling of $\alpha$ in Theorem 3 w.r.t. $(\Delta, k, n)$ may vary widely. For example, in the non-overlapping case, $\alpha = k_{\min}/n \leq 1/k$, where $k_{\min}$ is the size of the smallest cluster. We can add this and other examples in the revised version. In the worst-case, a positive $\alpha$ could be as small as $1/n$ (unreasonable in real datasets), which implies a query complexity of $O(n^2)$. This is much higher than our average case results, as expected. We will add a detailed discussion on the comparison.

**5)** More experiments: We have recently conducted a few more experiments on "Delicious Bookmarks" and "Last.FM" datasets from `grouplens.org`, which further confirm our conclusions. We can add these results in the revised paper.

**Reviewer #3**: **1)** Tightness of Bounds: Please see the response to Rev. 2. In the revised version, we will add a table summarizing the scaling of these bounds.

**2)** Dependency on $\Delta$: The dependency of the result in Theorem 5 on $\Delta$ is hidden in $\alpha$, which may vary widely in $(\Delta, k, n)$. Note that $\alpha$ decreases as a function of $\Delta$, which implies that the query complexity increases with $\Delta$. For example, consider the example of 3 equally-sized clusters $A$, $B$ and $C$. Suppose $\Delta = 1$ and in that case $|A \setminus B \cup C| = |A| = n/3$, implying that $\alpha = 1/3$. Now suppose that $\Delta = 2$. In this case $A \cap B$ and $A \cap C$ are non-empty and therefore $|A \setminus B \cup C| = |A| - |A \cap B| - |A \cap C| < n/3$, namely, $\alpha$ is less than $1/3$. As mentioned above, we plan to add several examples which will further clarify the behaviour of $\alpha$.

**3)** Non-overlapping case: In the non-overlapping case, $\alpha = k_{\min}/n \leq 1/k$, where $k_{\min}$ is the size of the smallest cluster. This implies that the query complexity in the best scenario is $O(nk \log n)$, and thus there is no contradiction.

**4)** Generative models: Indeed, for the generative models case, the exponential dependency of the upper bounds on $\Delta$ is inherent, as the information-theoretic lower bounds suggest, and so it is neither an artifact of the proposed algorithm nor the way we upper bound its performance.

**5)** Role of $|\mathcal{S}|$: The theoretical value of $T = |\mathcal{S}|$ in Theorem 5 does work. But in practice we can take even smaller values for $T$ than this theoretical value, and still guarantee recovery. This makes sense because Theorem 5 gives a sufficient condition on $T$. Of course, the smaller the size of $\mathcal{S}$ is, the sample & comp. complexities are smaller as well.

**6)** Unquantized responses: Algorithm 2 (`FindSimilarity`) is designed so that the guarantees hold under a specific stochastic assumption. More concisely, the necessary size of $\mathcal{S}$ is not defined for arbitrary real-world datasets. Note however, that the main objective in the first part of the algorithm is to select a number of elements so that the gram matrix is of full rank. Therefore, for a real-world dataset, we can always sample #clusters ($k$) elements randomly and make all pairwise queries and subsequently check if the gram matrix is of full rank. If it is not, we discard the elements and sample again, until we succeed. This is the reason why we chose 5 movies in the experiments (recall that the number of clusters is 5). Most real-world datasets are well behaved so that the number of trials required is actually quite low. We will add this discussion in the final version.

[Meta-Review · NeurIPS 2019]

The paper proposes simple methods and theoretical analysis of those for overlapping clustering using a few pairwise queries. The reviewers all agree that this is an interesting and significant problem. The reviewers also feel that their questions were well clarified in the rebuttal. The authors should make sure that these clarifications are added to the final version of the paper.